# Multi-model assessment of the atmospheric and radiative effects of supersonic transport aircraft

Jurriaan A. van 't Hoff[1], Didier Hauglustaine[2], Johannes Pletzer[3], Agnieszka Skowron[4], Volker Grewe[1,3], Sigrun Matthes[3], Maximilian M. Meuser[5], Robin N. Thor[3], Irene C. Dedoussi[1,6]

[1]Aircraft Noise and Climate Effects, Delft University of Technology, Delft, 2629 HS, The Netherlands

[2]Laboratoire des Sciences du Climat et de l'Environnement (LSCE), CEA-CNRS-UVSQ, Gif-sur-Yvette, 91190, France

[3]Deutsches Zentrum für Luft- und Raumfahrt, Institut für Physik der Atmosphäre, Oberpfaffenhofen, 82234 Weßling, Germany

[4]Faculty of Science and Engineering, Manchester Metropolitan University, Manchester, M1 5GD, United Kingdom

[5]Deutsches Zentrum für Luft- und Raumfahrt, Institute of Air Transport, 21079 Hamburg, Germany

[6]Department of Engineering, University of Cambridge, Cambridge, CB3 0DY, United Kingdom

*Correspondence to*: icd23@cam.ac.uk

**Abstract.** Commercial supersonic aircraft may return in the near future, offering reduced travel time while flying higher in the atmosphere than subsonic aircraft, and thus displacing part of the passenger traffic and associated emissions to higher altitudes. For the first time since 2007, we present a comprehensive multi-model assessment of the atmospheric and radiative effect of this displacement. We use four models (EMAC, GEOS-Chem, LMDZ-INCA, MOZART-3) to evaluate three scenarios where subsonic aviation is partially replaced with supersonic aircraft. Replacing 4% of subsonic traffic with a Mach 2 aircraft with a $NO_x$ emissions index of 13.8 g($NO_2$)/kg leads to ozone column loss of -0.3% (-0.9 DU, model range -0.4% to -0.1%), and it increases radiative forcing by 19.1 mW/m$^2$ (model range 16.7 to 28.1). This forcing is driven by water vapour (18.2 mW/m$^2$), ozone (11.4 mW/m$^2$), and aerosol emissions (-10.5 mW/m$^2$). The use of a Mach 2 concept with low $NO_x$ emissions (4.6 g($NO_2$)/kg) reduces the effect on forcing and ozone to 13.4 mW/m$^2$ (model range 2.4 to 23.4) and -0.1% (-0.3 DU, model range -0.2% to +0.0%), respectively. If a Mach 1.6 aircraft with lower cruise altitude and $NO_x$ emissions of 4.6 g($NO_2$)/kg is used instead, we find a net-zero effect on the ozone column and an increase in radiative forcing of 3.7 mW/m$^2$ (model range 0.5 to 7.1). The supersonic concepts have up to 185% greater radiative effect per passenger kilometre from non-$CO_2$ emissions compared to subsonic aviation.

## 1 Introduction

Over the past decades there has been growing global demand for fast intercontinental transportation. This demand has led to a search for faster alternatives to subsonic aircraft, such as supersonic or even hypersonic vehicles (Kinnison et al. 2020; Pletzer et al. 2022; Matthes et al. 2022; Eastham et al. 2022). Supersonic transport aircraft (SSTs) have already attracted considerable commercial interest and several parties are working towards the reintroduction of civil supersonic transportation, relying on

new technologies such as low-boom hull designs to minimize the environmental effect of the sonic boom (Berton et al. 2020). An example of this development is NASA's X-59 demonstrator aircraft, which has recently gone through engine testing and is planned to have its first flight in 2025.

SSTs generally use higher cruise altitudes to mitigate drag, ranging from 14 to 21 km compared to subsonic aircraft which typically cruise between 9 and 12 km. The increase in operational altitude changes the atmospheric response to the aircraft's emissions. Of particular concern are the emissions of nitrogen oxides ($NO_x$), water vapour ($H_2O$), and sulphur compounds, which affect the distribution and chemistry of ozone ($O_3$) and global radiative forcing (RF). $NO_x$ and $H_2O$ emissions from SSTs lead to catalytic destruction of ozone through the $NO_x$ and $HO_x$ cycles (Matthes et al. 2022; Grewe et al. 2007; Solomon

1999; Crutzen 1972; Johnston 1971), while the latter leads to the formation of sulphate aerosols ($SO_4$) that facilitate ozone destruction through heterogeneous chemistry (Pitari et al. 2014; Brasseur and Granier 1992). Through these emissions, the adoption of SSTs has been previously been linked to large-scale changes in the ozone distribution, with higher emission altitudes being linked to increased depletion of the global ozone column (van 't Hoff et al., 2024a; Fritz et al. 2022; Zhang et al. 2021a; Speth et al. 2021). This is associated with a risk to public health, as the subsequent increase in surface UV-exposure

affects mortality (Eastham et al., 2018). Additionally, several studies have identified the changes in the ozone distribution as the primary warming driver of the radiative effect of non-$CO_2$ emissions from supersonic aircraft (van 't Hoff et al. 2024a; Zhang et al. 2023; Eastham et al. 2022), although others have also found this to have a net-cooling effect instead (Zhang et al. 2021b; Grewe et al. 2007).

Other non-$CO_2$ emissions that affect RF are water vapour and aerosols (black carbon, sulphate). Water vapour directly affects RF and it plays a pivotal role in the climate effect of subsonic aviation through the formation of contrails (Lee et al. 2021). At supersonic cruise altitudes contrail formation is expected to be much less common due to the drier conditions in the stratosphere (Stenke et al. 2008; Grewe et al 2007; IPCC 1999), however, the water vapour perturbation lifetime is higher compared to subsonic altitudes. Grewe and Stenke (2008) estimate that this lifetime is up to around 1.5 years at 20 km altitude, compared

to lifetimes of 1 to 6 months at subsonic cruise altitudes. This facilitates more accumulation of stratospheric water vapour, which has a direct warming effect. The emission of water vapour has also been identified as a critical, if not the primary, driver of the radiative effect of SSTs and hypersonic vehicles (Pletzer and Grewe 2024; Pletzer et al. 2022; Zhang et al. 2021a; Grewe et al. 2010, 2007). The stratospheric accumulation of aerosols also affects RF, but they are commonly associated with a cooling effect instead (van 't Hoff 2024a; Zhang et al. 2023; Eastham et al. 2022; Speth et al. 2021). Combined, most studies find that

these emissions result in net-warming RF in response to the adoption of supersonic aircraft (van 't Hoff 2024a; Zhang et al 2023,2021a; Eastham et al. 2022; Speth et al. 2021; Grewe et al. 2007).

Both the ozone and climate effects stem from changes in the chemical composition of the atmosphere, particularly the stratosphere. To adequately capture these changes we rely on chemistry transport models (CTMs) or climate chemistry models

(CCMs), which model the chemistry, transport, removal, and conversion of species throughout the atmosphere. A variety of such models has already been used to evaluate the effects of high-altitude emissions on ozone and RF, and despite their similar scope and chemistry routines, these models often yield different results. These differences are partially driven by uncertainties about the future of supersonic civil aviation, resulting in the use of different emission scenarios and timelines across studies. However, multi-model studies also highlight that there are considerable differences between different CTMs and CCMs even

in the evaluation of identical scenarios, which has been reported in studies of both supersonic (Pitari et al. 2008; Grewe et al., 2007; Kawa et al. 1999) and subsonic aviation (Olsen et al. 2013). These differences are most prevalent in the evaluation of the ozone response, which is subject to complex feedback mechanisms. Differences in the modelling thereof can result in a large spread in model predictions of the ozone response and its effect on RF, at times leading to contradictory results between models (e.g., Grewe et al. 2007; Kawa et al. 1999). The effect of these differences is also evident when metrics such as

sensitivities to specific emission species are compared between studies and models (van 't Hoff et al. 2024a; Eastham et al. 2022).

The differences in model responses to SST emissions are often driven by different implementations of chemical, transport, or radiative processes across the models, or by differences in interactions between these model components. They are also

affected by fundamental properties, such as the model resolution. Understanding the effect of model-driven differences is vital to our capability to synthesize results across studies that use different models. Multi-model studies expose these differences, and can help us understand their drivers, potentially offering robust conclusions and policy advice. In the field of SST emissions the most recent multi-model study performed was by Grewe et al. (2007). They showed that the four atmospheric models they used agreed that the introduction of SST emissions led to net depletion of ozone, accumulation of stratospheric

water vapour, and a net-warming radiative effect. However, they also showed that there was a spread in the calculations of these effects, both in terms of the spatial distribution and in absolute numbers. For example, they report a model-mean ozone perturbation of -8 Tg with a range of -16 to -1 Tg and a standard deviation of 5.5 Tg. Before that, Kawa et al. (1999) used seven different models to study the effect of SST emissions, reporting a similar spread in model calculations (e.g., they report a mean ozone column loss of -0.17% with a range of -0.6 to 0.23% and a standard deviation of 0.22%, HSR scenario 4). In

both cases the standard deviation of the model predictions is similar to the mean, highlighting the magnitude of model-driven differences.

Over the past decades there have been considerable advances in our understanding of the underlying chemistry and physics, and at the same time the increased availability of computational power has expanded our capacity to model these processes.

This has led to enhancements in the overall modelling capabilities of CTMs and CCMs. To assess the effect of these developments Zhang et al. (2021a) have compared the WACCM6 model to models used by Kawa et al. (1999) in a reassessment of their scenarios, finding similar overall atmospheric effects despite the higher fidelity in their model. While this does provide some insight with respect to older evaluations, it remains unclear how the past two decades of model

development affect model-driven differences in assessments of an identical scenario. Understanding these differences can help us better synthesize results from studies that use different models.

To close this gap, we present a comprehensive study of the effect of the partial replacement of subsonic traffic with SSTs on atmospheric composition and RF, using four widely-used chemistry climate and chemistry transport models (EMAC, GEOS-Chem, LMDZ-INCA, MOZART-3). We evaluate three SST adoption scenarios based on the scenarios considered by Grewe et al. (2007), which reflect the partial replacement of subsonic traffic with different supersonic aircraft concepts or emission characteristics. We also analyse differences in atmospheric responses between the models. In particular, we cover the responses of water vapour, $NO_x$, ozone, odd oxygen loss rates, and RF, as well as how these differ between the models. The output of these models is presented in a harmonized way, in order to provide a comprehensive, multi-model, overview of the atmospheric and radiative effects of the adoption of supersonic aircraft.

## 2. Emission scenarios

Our emission scenarios are based on emission scenarios from the SCENIC project (Grewe and Stenke 2008; Grewe et al. 2007). The SCENIC emission scenarios consider the adoption of a fleet of SSTs in 2050, replacing part of the revenue passenger kilometres (RPK) of subsonic aviation. We consider a baseline scenario (S0) with only subsonic aviation emissions (also S0 from Grewe et al. 2007). The nominal supersonic scenario (S1) considers the replacement of 4% of subsonic RPK with a fleet of 501 SSTs, operating at Mach 2.0 with cruise altitudes of 16.5 to 19.6 km (S5 of Grewe et al. 2007). This results in an increase of 6.3% in global aviation fuel usage. The triple $NO_x$ scenario (S2) is a variant of the nominal scenario (S1) with tripled supersonic $NO_x$ emissions, resulting in a fleet-average emission index of 13.80 kg $NO_2$/kg for the SSTs. This is closer to $NO_x$ emission indices from recent SST concepts (Zhang et al. 2023, 2021a; Fritz et al. 2022; Eastham et al. 2022; Speth et al. 2021). In the low cruise scenario (S3) we consider the use of a SST with a lower cruise altitude of 13.1 to 16.7 km, and a cruise speed of Mach 1.6 (P6 from Grewe et al. 2007). Compared to the nominal emissions, this leads to a 5.5% reduction in supersonic RPK and a 31% reduction in SST fuel consumption. The characteristics of the emission scenarios are summarized in Table 1, and the resulting changes in the distribution of aviation emissions are shown in Figure 1.

**Table 1: Summary of the sub- and supersonic aircraft emissions in the supersonic scenarios. The baseline scenario (S0) has no supersonic aviation, and there are three scenarios considering the partial replacement of subsonic aviation with supersonic aircraft. These are denoted as the nominal supersonic scenario (S1), a triple $NO_x$ scenario (S2), and the low cruise scenario (S3). In all of the supersonic scenarios subsonic traffic is partially replaced by the supersonic aircraft, reducing the fuel consumption of subsonic aircraft compared to the baseline. Within each category, the left (Sub.) column summarizes the subsonic aircraft emissions, and the right (Sup.) column summarizes the supersonic aircraft emissions.**

| Scenario | | RPK | | Fuel consumption | | Avg. EI NOx | | NOx emissions | | Cruise altitude | |
| --- | --- | --- | --- | --- | --- | --- | --- | --- | --- | --- | --- |
| | | $(10^{11}$ px. km) | | (Tg yr$^{-1}$) | | (g (NO$_2$) kg$^{-1}$) | | (Tg(NO$_2$) yr$^{-1}$) | | (km) | |
| | | Sub. | Sup. | Sub. | Sup. | Sub. | Sup. | Sub. | Sup. | Sub. | Sup. |
| **Baseline** | **(S0)** | 178.2 | - | 656.4 | - | 10.91 | - | 7.16 | - | 9-13 | - |
| **Nominal** | **(S1)** | 171.1 | 7.3 | 639.9 | 57.9 | 10.91 | 4.60 | 6.98 | 0.27 | 9-13 | 16.5-19.5 |
| **Triple NOx** | **(S2)** | 171.1 | 7.3 | 639.9 | 57.9 | 10.91 | 13.80 | 6.98 | 0.80 | 9-13 | 16.5-19.5 |
| **Low cruise** | **(S3)** | 171.5 | 6.9 | 639.0 | 40.0 | 10.84 | 5.62 | 6.93 | 0.22 | 9-13 | 13.1-16.7 |

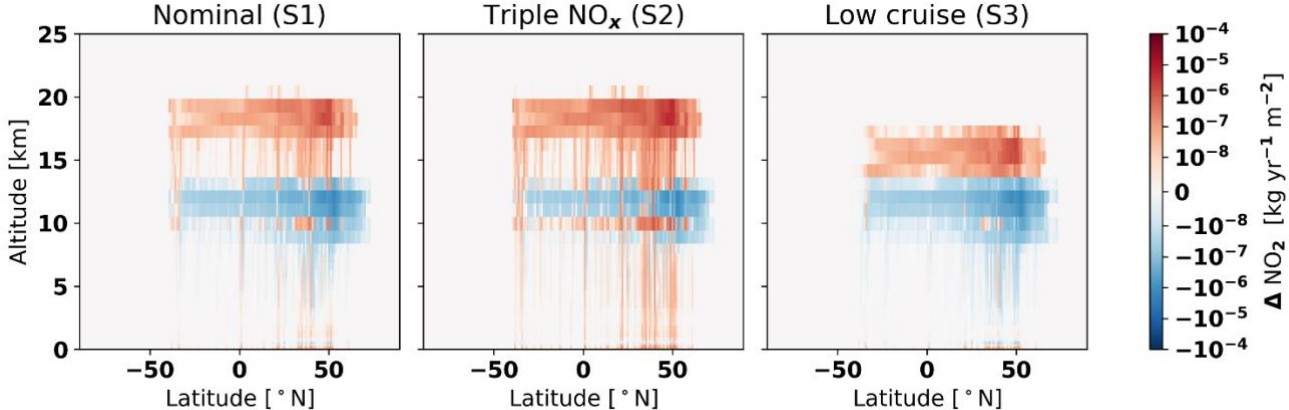

**Figure 1: Zonal mean changes in the distribution of annual NOx emissions (expressed in kg NO₂ m⁻² yr⁻¹) due to the partial replacement of subsonic traffic with SSTs. Differences are calculated with respect to the annual baseline (S0) emissions.**

3 Atmospheric Modelling

We evaluate the effect of the changes in aviation emissions on atmospheric composition and radiative forcing using four widely used chemistry transport models; EMAC, GEOS-Chem, LMDZ-INCA, and MOZART-3. Key characteristics of these models, including the horizontal and vertical resolution, chemistry processes, and dynamics, are summarized in Table 2. A direct comparison of the vertical grid of the models is also shown in Figure A1. We evaluate the effect of the SST adoption on a future atmosphere based on projections of the atmospheric composition and anthropogenic emissions in 2050, although there are some differences in how this is incorporated in the different models given model input availability and other technical restrictions. The next subsections discuss the technical details and setup of each model individually.

**Table 2: Summary of the atmospheric models characteristics, including resolution, chemistry, and dynamics.**

| Model | Resolution (lat × lon) | Vertical domain & resolution | Chemistry | Dynamics | Reference |
|---|---|---|---|---|---|
| EMAC | T42 (~2.8°×2.8°) | Surface to 0.01 hPa, 90 hybrid levels<br><br>22 layers between 400 and 50 hPa | 1202 species<br>1839 gas phase rcts.<br>401 aqueous phase rcts.<br>401 photolytic rcts.<br>27 aqueous phase photolytic rcts.<br>21 heterogeneous rcts. | ECHAM5 nudged to ERA5, coupled online meteorology | Jöckel et al. 2016<br>Roeckner et al. 2003<br>Sander et al. 2011 |
| GEOS-Chem | C48 (~2°×2.5°) | Surface to 0.01 hPa,<br>72 hybrid levels.<br><br>14 layers between 400 and 50 hPa | 132 species<br>344 kinetic rcts.<br>154 photolytic rcts.<br>78 heterogeneous rcts. | MERRA -2, offline meteorology | Eastham et al. 2018<br>Eastham et al. 2014<br>Bey et al. 2001 |
| LMDZ-INCA | 1.3°×2.5° | Surface to 0.04 hPa, 39 hybrid levels<br><br>11 layers between 400 and 50 hPa | 154 species<br>234 homogeneous rcts.<br>43 photolytic rcts.<br>30 heterogeneous rcts. | LMDZ nudged to ERA5, offline meteorology | Hauglustaine et al. 2014.<br>Terrenoire et al. 2022. |
| MOZART-3 | T42 (~2.8°×2.8°) | Surface to 0.1 hPa, 60 hybrid layers<br><br>15 layers between 50 and 400 Hpa | 108 species<br>218 gas phase rcts.<br>71 photolytic rcts.<br>18 heterogeneous rcts. | ERA-Interim, offline meteorology | Kinnison et al., 2007<br>Skowron et al., 2021 |

## 3.1 EMAC

EMAC is an atmospheric chemistry general circulation model consisting of the dynamical core ECHAM5 (European Centre HAMburg general circulation model, version 5, Roeckner et al. 2006) and MESSy (Modular Earth Submodel System, Jöckel et al. 2016). The chemical mechanism incorporates 1839 gas phase and 21 heterogeneous phase reactions between 1202 species, which includes type 1ab and type 2 polar stratospheric cloud (PSC) processes (Kirner et al. 2011; Jöckel et al. 2010). The reaction rates are from the most recent Jet Propulsion Laboratory evaluation, number 19 (Burkholder et al. 2020). Aerosol background concentrations of sulphates are provided for heterogeneous chemistry using inventories prepared for the Chemistry Climate Model Initiative (Jöckel et al. 2016; Gottschaldt et al. 2013). Water vapour is accounted for as specific humidity, which is influenced by gas-, solid- and liquid-phase processes at all altitudes. It is produced through 55 reactions and destroyed through six reactions, and it is affected by physical processes such as rain-out and sedimentation. The radiation scheme incorporates 81 bands and recreates the solar cycle with high fidelity. This applies to the region from the top of the model

domain (0.01 hPa) to 70 hPa (Kunze et al. 2014; Dietmüller et al. 2016). RF is assessed at the tropopause with the radiative code of ECHAM5 (Roeckner et al. 2006), as well as a new radiative code based on the work by Pincus and Stevens (2013), as implemented by Nützel et al. (2024).

In this work we use EMAC version 2.55.2 (The MESSY Consortium, 2021) with a T42 global grid (approximately $2.8° \times 2.8°$ latitude, longitude) and 90 hybrid vertical levels from the surface up to 80 km. Twenty-two of these layers are located between 400 and 50 hPa, with an average thickness of 0.6 km. The model has online meteorology which is nudged towards ERA5 reanalysis data (2000-2010) between the surface and 10 hPa. Nudging is applied in the same way as earlier studies (Pletzer et al. 2022; Jöckel et al. 2016), affecting horizontal and vertical winds, temperature (wave 0 omitted), and the logarithm of surface pressure. The background atmosphere, surface boundary conditions, and non-aviation anthropogenic emissions are based on the CMIP6 SSP3-7.0 scenario for the year 2050 (Meinshausen et al. 2020). Volcanic emissions are included based on the AEROCOM emission inventory (Dentener et al. 2006; Ganzeveld et al. 2006) for the year 2000, which is cycled throughout the model run. We also apply a spin up method to reduce spin up times while maintaining annual quasi-equilibrium. This is done by applying an altitude-dependent scaling factor to the emissions during the first year of the model run, so that the annual quasi-equilibrium is achieved faster. For more detail on this method, we refer to the work by Pletzer and Grewe (2024) and its supplement. The model is ran for a total of 16 years to allow for a longer analysis period and better statistical significance of the results.

**3.2 GEOS-Chem**

GEOS-Chem is a community-developed tropospheric-stratospheric CTM with over 280 chemical species based on the Goddard Earth Observing System (GEOS) (Bey et al. 2001). The model uses KPP for kinetic chemistry (Damian et al. 2002) and Fast-JX for photolytic reactions (Bian and Prather, 2002), incorporating stratospheric chemistry through the Unified Tropospheric-Stratospheric Chemistry Extension (UCX) by Eastham et al. (2014). Within the troposphere, water vapour mixing ratios are prescribed by meteorology, and in the stratosphere the water vapour tracer evolves freely subject to gas-phase chemistry, photochemistry, and transport. GEOS-Chem's capability to model stratospheric chemistry has been demonstrated against satellite observations in several studies (Fritz et al. 2022; Speth et al. 2021; Eastham et al. 2014), and it is incorporated into NASA GMAO's GEOS chemical composition forecast (GEOS-CF) (Keller et al. 2021). The model simulates the distribution of various aerosols from anthropogenic and natural sources, and it models heterogeneous reactions in both the tropo- and stratospheric domain, including the formation, sedimentation and evaporation of PSCs (Eastham et al. 2014). RF is evaluated at the tropopause in the same manner as described in van 't Hoff et al. (2024a), incorporating stratospheric adjustment following the implementation by Eastham et al. (2022).

We use version 14.1.1 of the GEOS-Chem High Performance (GCHP) model (The International GEOS-Chem User Community, 2023) with a C48 cubic-spherical global grid (approximately $2° \times 2.5°$ latitude, longitude) and 72 non-uniform

vertical levels. The vertical grid has 14 layers between 400 and 50 hPa, with an average thickness of 0.9 km. We use historical meteorological data for the years 2000 to 2010 from the MERRA-2 reanalysis product by NASA/GMAO (Gelaro et al. 2017). Volcanic emissions are incorporated through historical emissions for the same time period following work by Carn et al. (2015). Surface emissions and mixing ratios of long-lived species are prescribed following the 2050 boundary conditions of the SSP3-7.0 scenario. Simulations are ran for a total of 10 years using the same spin up method as EMAC, detailed in Pletzer and Grewe (2024).

## 3.3 LMDZ-INCA

The LMDZ-INCA global chemistry-aerosol-climate model couples the LMDZ (Laboratoire de Météorologie Dynamique, version 6) General Circulation Model (GCM, Hourdin et al., 2020) and the INCA (INteraction with Chemistry and Aerosols, version 6) model (Hauglustaine et al., 2014; 2004). LMDZ-INCA is part of the IPSL Coupled Model, and we use the "Standard Physics" parameterization of the GCM (Boucher et al., 2020). The large-scale advection of tracers is calculated based on a monotonic finite-volume second-order scheme (Van Leer, 1977; Hourdin and Armengaud 1999). Deep convection is parameterized according to the scheme of Emanuel (1991). The turbulent mixing in the planetary boundary layer is based on a local second-order closure formalism. INCA includes state-of-the-art $CH_4$-$NO_x$-CO-NMHC-$O_3$ tropospheric photochemistry (Folberth et al., 2006); Hauglustaine et al., 2004) as well as interactive chemistry in the stratosphere and mesosphere (Terrenoire et al., 2022). The INCA model simulates the distribution of aerosols with anthropogenic sources such as sulphates, nitrates, black carbon (BC), and organic carbon, as well as natural aerosols such as sea-salt and dust. Both natural and anthropogenic tropospheric aerosols facilitate heterogeneous reactions (Hauglustaine et al., 2014, 2004). Heterogeneous processes on PSCs and stratospheric aerosols are parameterized following the scheme implemented in Lefèvre et al. (1994). INCA incorporates a water vapour tracer which is linked to the LMDZ GCM. Similar to GEOS-Chem, this tracer is prescribed by LMDZ below the tropopause, and it evolves freely in the stratosphere subject to chemistry (gas-phase and photochemical), transport, condensation, sedimentation, and stratospheric emissions. RF is evaluated using an improved version of the ECMWF scheme developed by Fouquart and Bonnel (1980) in the solar part of the spectrum and by Morcrette (1991) in the thermal infrared. Aerosol forcing is assessed at the top of the atmosphere, similar to Hauglustaine et al. (2014), and forcing from ozone and water vapour is calculated at the tropopause with an offline version of the LMDZ GCM with stratospheric adjustment, similar to Terrenoire et al. (2022).

We use a configuration with a horizontal resolution of 1.3° × 2.5° in latitude and longitude, with 39 hybrid vertical levels extending up to 70 km. Eleven of these layers are located between 300 and 50 hPa, with an average thickness of 1.1 km. The model is ran for 15 years, with initial conditions representative of the year 2050 (Pletzer et al., 2022). Surface emissions and boundary conditions for 2050 are prescribed by the CMIP6 SSP3-7.0 scenario (Meinshausen et al. 2020). Stratospheric volcanic aerosols are based historical data (2000-2014) from Input4MIP for the calculation of heterogeneous chemistry. In this study, the LMDZ GCM zonal and meridional wind components are nudged towards the meteorological data from the

European Centre for Medium-Range Weather Forecasts (ECMWF) ERA-Interim reanalysis, with a relaxation time of 3.6 h (Hauglustaine et al., 2004). The ECMWF fields are provided every 6 h and interpolated onto the GCM grid for the years 2004-2018.

## 3.4 MOZART-3

The Model for OZone And Related chemical Tracers, version 3 (MOZART-3) is an offline CTM (Kinnison et al., 2007) that has been used for an extensive range of applications, including various aspects of the effect of aircraft $NO_x$ emissions on atmospheric composition (e.g. Skowron et al., 2021, 2015, 2013; Freeman et al., 2018; Sovde et al., 2014; Flemming et al., 2011; Liu et al., 2009). MOZART-3 accounts for advection based on a flux-form semi-Lagrangian scheme, a shallow and mid-level convective and deep convective routines, boundary layer exchanges, and wet and dry deposition. MOZART-3 reproduces detailed chemical and physical processes from the troposphere through the stratosphere, including gas-phase, photolytic, and heterogeneous reactions. The latter includes four aerosol types: liquid binary sulphate, supercooled ternary solution, nitric acid tri-hydrate, and water-ice. Heterogeneous processes occurring on liquid sulphate aerosols and PSCs are also included, following the approach of Considine et al. (2000). The kinetic and photochemical data are based on the NASA/JPL evaluation (Sander et al., 2006). Water vapour tracers have been implemented into the model for the purpose of this work, allowing water vapour to evolve freely in the stratosphere subject to transport and chemistry. We assess RF at the tropopause using the SOCRATES model of the UK Met Office (Manners et al. 2015).

We use a model configuration with a T42 (~ 2.8° × 2.8°) horizontal resolution and 60 hybrid layers from the surface to 0.1 hPa. The vertical grid has 15 layers between 400 and 50 hPa, with an average thickness of 0.8 km. The transport of chemical compounds is driven by 6 hour reanalysis ERA-Interim data for the year 2006 from the European Centre for Medium Range Weather Forecast (ECMWF). The 2050 gridded surface emissions (anthropogenic and biomass burning) are prescribed by Integrated Assessment Models (IAMs) for the business-as-usual scenario of the Representative Concentration Pathways, RCP 4.5. The surface boundary conditions for long-lived species are set to fixed volume mixing ratio units with their concentrations determined using the methodology of Meinshausen et al. (2011). This future scenario does not include natural emissions, such as isoprene, $NO_x$ from lightning and soil, or oceanic emissions of CO. The model is integrated for 8 years until a steady-state is reached, and the last year of these simulations is considered for the analysis. The assessment of the water vapour perturbation is performed using separate model runs, the output of which has a limited vertical resolution with 30 layers from 200 to 0.1 hPa.

## 3.5 Approach for evaluating atmospheric and radiative effects

We quantify the effect of the supersonic emissions by comparing the perturbed atmospheric composition and forcing of the supersonic scenarios with that of the baseline simulation, thereby also taking into account the effects of the reduction in subsonic emissions. To account for inter-annual variability, we calculate the effect of the emissions over the last three years

of the model integrations for GEOS-Chem and LMDZ-INCA. For EMAC we average over 6 years to improve the statistical significance of the results, considering the added variability from its online meteorology. For MOZART-3 we show an annual average considering its cycling meteorology. We calculate the stratospheric perturbation lifetime (e-folding lifetime) of emission species by dividing the stabilized stratospheric perturbation by the increase in annual stratospheric emissions. Since

not all models calculate forcing from aerosol perturbations, we first calculate model-mean RF from ozone, water vapour, and aerosols separately, which are then combined to produce a first-order estimate of the net radiative effect.

## 4 Results

We present a comprehensive review of the effects of the partial replacement of subsonic traffic with SSTs on atmospheric composition and RF using four atmospheric chemistry transport models for the first time since the work by Grewe et al. (2007).

In section 4.1 we summarize the model-mean (mean over all models) effect of the adoption of the SST fleets on the atmospheric composition and RF, and we compare the models' baseline atmospheres. Sections 4.2 to 4.6 discuss in more detail how the supersonic scenarios affect stratospheric water vapour, nitrogen oxides, ozone, odd oxygen ($O_x$) loss rates, and RF, respectively. In these sections we also explore the differences between the models we use.

### 4.1 Global atmospheric and radiative effect

Table 3 provides a summary of the key variables representing the changes in atmospheric composition and RF in response to the supersonic scenarios across all models. Comprehensive tables of the effects on water vapour, $NO_x$, and ozone, are included in the appendix (Tables A1 to A3). Similar to Grewe et al. (2007), we include the hemispheric ratio, which is the ratio of the perturbation mass in the northern hemisphere over the perturbation mass in the southern hemisphere, as a means to quantify the mixing of emissions between hemispheres. For reference, the hemispheric ratio for the SST fuel consumption is 10.14 for

the nominal and triple $NO_x$ scenarios, and 10.34 for the low cruise scenario, indicating the that the vast majority of SST emissions take place in the northern hemisphere.

In response to the nominal supersonic scenario (S1), we find a model-mean stratospheric water vapour perturbation of 46.9 Tg (model range 20.1 to 63.3 Tg) with a lifetime of 12.0 months (model range 5.2 to 16.2). The change in aviation emissions

leads to increases in stratospheric $NO_x$, with a model-mean perturbation of 38.9 Gg($NO_2$) (model range 32.1 to 43.5), and global ozone column changes of -0.1% (-0.3 DU, model range -0.2% (-0.7 DU) to (0.0% (0.0 DU)). RF is also affected, with the largest forcing being from water vapour (20.8 mW/m$^2$, model range 6.2 to 32.3), followed by ozone (3.2 mW/m$^2$, model range 1.3 to 6.8). Increases in stratospheric aerosols have a cooling effect, with forcing of -0.4 mW/m$^2$ from black carbon and -9.7 mW/m$^2$ from inorganic aerosols (sulphates & nitrates). We therefore estimate a model-mean net RF of 13.9 mW/m$^2$ when

aerosols are included (model range 2.9 to 24.4).

**Table 3: Summary of effects on stratospheric water vapour, stratospheric NOx, ozone column, and RF for the SST scenarios. These values are calculated as differences between the perturbed and baseline atmospheres. For more extensive summaries of the effects on H₂O, NOₓ, and O₃, including background mass budgets, see Tables A1-A3 in the appendix. The inorg. aer. column contains the RF from changes in nitrates and sulphates.**

Notes: [a] For EMAC two numbers are shown for the RF assessment: the upper is calculated using the ECHAM5 radiative scheme, the bottom with the scheme by Pincus and Stevens (2013). Both are considered in the mean. [b] These aerosol forcings are calculated at the top of the atmosphere. [c] Total forcing with aerosols is calculated with the model-mean aerosol forcings.

| | Stratospheric H₂O | | | Stratospheric NOₓ | | O₃ column | Radiative Forcing | | | | | |
| --- | --- | --- | --- | --- | --- | --- | --- | --- | --- | --- | --- | --- |
| | Perturbation | Perturbation lifetime | Hemispheric ratio (increase only) | Perturbation | Perturbation lifetime | Perturbation | O₃ | H₂O | Total (O₃+H₂O) | BC | Inorg. Aer. | Total[c] |
| **Nominal (S1)** | [Tg] (%) | [months] | [NH/SH] | [Tg NO₂] (%) | [months] | [DU] (%) | | | [mW/m²] | | | |
| EMAC | 63.3 (1.5 %) | 16.2 | 4.0 | 37.4 (1.6 %) | 4.0 | 0.0 (0.0 %) | 2.8[a] 3.0[a] | 29.7[a] 22.2[a] | 32.5[a] 25.2[a] | - | - | 22.4 15.2 |
| GEOS-Chem | 49.3 (0.7 %) | 12.7 | 4.0 | 43.5 (1.9 %) | 4.7 | -0.7 (-0.2 %) | 1.3 | 13.4 | 14.7 | -1.3 | -9.3 | 4.6 |
| LMDZ-INCA | 20.1 (0.6%) | 5.2 | 5.4 | 42.6 (1.7 %) | 4.6 | -0.2 (-0.0 %) | 6.8 | 6.2 | 13.0 | 0.5[b] | -10.0[b] | 2.9 |
| MOZART-3 | 54.7 (1.6 %) | 14.0 | 4.6 | 32.1 (1.6 %) | 3.5 | -0.6 (-0.2%) | 2.2 | 32.3 | 34.5 | - | - | 24.4 |
| Model-mean | 46.9 (1.1 %) | 12.0 | 4.5 | 38.9 (1.7%) | 4.2 | -0.3 (-0.1 %) | 3.2 | 20.8 | 24.0 | -0.4 | -9.7 | 13.9 |
| **Triple NOₓ (S2)** | | | | | | | | | | | | |
| EMAC | 61.8 (1.5 %) | 15.8 | 4.1 | 140.5 (6.0 %) | 3.3 | -0.6 (-0.2 %) | 7.5[a] 6.9[a] | 31.1[a] 21.3[a] | 38.6 28.2 | - | - | 28.4 18.0 |
| GEOS-Chem | 49.3 (0.7 %) | 12.8 | 3.9 | 173.6 (7.5 %) | 4.1 | -1.4 (-0.4%) | 13.3 | 14.0 | 27.3 | -1.3 | -9.2 | 17.1 |
| LMDZ-INCA | 20.6 (0.6 %) | 5.3 | 5.2 | 119.5 (4.9 %) | 2.8 | -0.3 (-0.1 %) | 20.9 | 6.3 | 27.2 | 0.5[b] | -10.3[b] | 17.0 |
| MOZART-3 | - | - | - | 112.8 (5.5 %) | 2.6 | -1.4 (-0.4 %) | 8.6 | - | - | - | - | - |
| Model-mean | 44.1 (0.9 %) | 11.3 | 4.4 | 136.6 (6.0 %) | 3.2 | -0.9 (-0.3 %) | 11.4 | 18.2 | 29.6 | -0.4 | -9.8 | 19.4 |
| **Low cruise (S3)** | | | | | | | | | | | | |
| EMAC | 16.0 (0.4 %) | 9.1 | 3.1 | 18.2 (0.8 %) | 4.6 | 0.1 (0.0 %) | 2.4[a] 2.4[a] | 8.2[a] 6.3[a] | 10.6 8.7 | - | - | 7.2 5.3 |

| | | | | | | | | | | | | |
|---|---|---|---|---|---|---|---|---|---|---|---|---|
| GEOS-Chem | 6.0 (0.1 %) | 3.4 | 10.4 | 23.7 (1.0 %) | 6.0 | -0.0 (-0.0 %) | 2.1 | 1.9 | 4.0 | -0.4 | -3.1 | 0.6 |
| LMDZ-INCA | 2.4 (0.1 %) | 1.3 | 38.1 | 17.4 (0.7 %) | 4.4 | 0.1 (0.0 %) | 4.6 | 0.7 | 5.3 | 0.2[b] | -3.5[b] | 1.9 |
| Model-mean | 8.1 (0.2 %) | 4.6 | 17.2 | 19.8 (0.8 %) | 5.0 | 0.1 (0.0 %) | 2.9 | 4.3 | 7.2 | -0.1 | -3.3 | 3.8 |

In case of the triple $NO_x$ scenario (S2), the stratospheric $NO_x$ accumulation increases by a factor of 3.5 to a model-mean of 136.6 $Gg(NO_2)$ (model range 112.8 to 173.6). In this case the model-mean water vapour perturbation is 44.1 Tg (model range 20.6 to 61.8), and the model-mean ozone column depletion increases to -0.3% (-0.9 DU, model range -0.4% (-1.4 DU) to -0.1% (-0.3 DU)). RF from ozone is also enhanced, increasing to 11.4 $mW/m^2$ (model range 6.9 to 20.9), but RF from water vapour is still dominant with a mean value of 18.2 $mW/m^2$ (model range 6.3 to 31.1). We find RF of –0.4 $mW/m^2$ for black carbon and -9.8 $mW/m^2$ for inorganic aerosols. Including these, the estimated model-mean net RF is 19.4 $mW/m^2$ (model range 17.0 to 28.4).

When the supersonic cruise altitude and speed are reduced (scenario S3), the effects of the SST adoption on the atmospheric composition and RF are reduced as well. Scenario S3 has 30% less SST fuel burn compared to the nominal scenario (S1), but the reduction in atmospheric and radiative effects exceeds that. In this case we find a model-mean water vapour perturbation of 8.1 Tg (model range 2.4 to 16.0) with a lifetime of 4.6 months (model range 1.3 to 9.1). The stratospheric $NO_x$ perturbation is reduced to a model-mean of 19.8 $Gg(NO_2)$ (model range 17.4 to 23.7) and the ozone column changes by a global mean of 0.0% (0.1DU, model range 0.0% (0.0 DU) to 0.0% (0.1 DU)). The accumulation of stratospheric water vapour still has the largest contribution to radiative forcing (4.3 $mW/m^2$, model range 0.7 to 8.2), followed by cooling from aerosols (-0.1 $mW/m^2$ for black carbon and -3.3 $mW/m^2$ for inorganic aerosols) and ozone (2.9 $mW/m^2$, 0.7 to 4.6). The estimated total forcing is 3.8mW/m^2 (range 0.6 to 7.2).

Despite some differences in the model configurations and inventories, we find that the models have similar budgets of water vapour, $NO_x$, ozone, and halogens in their baseline atmospheres (Tables A1 to A4). The GEOS-Chem model stands out as having more stratospheric water vapour than the other models. Furthermore, MOZART-3's baseline atmosphere has around 15% less stratospheric $NO_x$ compared to the other models. This may be related to the use of RCP 4.5 boundary conditions rather than SSP3-7.0 (Meinshausen et al. 2020), and also to the use of ECMWF reanalysis meteorology, as this has been reported to lead to underestimations of stratospheric $NO_x$ mixing ratios before with the MOZART-3 model (Kinnison et al., 2007). The effects of the differences in baselines on the response to the SST emissions are discussed further in the relevant sections.

## 4.2 Water vapour

Figure 2 shows the zonal average water vapour perturbations from the nominal supersonic scenario (S1) as evaluated by the
four models. The vertical averages for all three scenarios are shown in Figure A2. We find that the perturbation patterns of
stratospheric water vapour agree across the models. The strongest increases, in terms of mixing ratios, occur around the cruise
altitude in the northern hemisphere, coinciding with the majority of SST emissions. From the cruise regions we see extensions
transporting water vapour to the northern polar latitudes, and upwards transport to the upper stratosphere in tropical latitudes.

Between the models we find a spread in the calculated water vapour perturbation lifetimes and hemispheric ratios, which is
indicative of differences in transport processes or chemical sinks between the models. Earlier works have identified that the
model resolution is important to the representation of transport, mixing, and diffusion processes (Revell et al. 2015; Roeckner
et al. 2006; Strahan and Polansky, 2006), and we also find a trend between the model grids and water vapour perturbation
lifetimes and hemispheric ratios (Figure 3). We find that the water vapour perturbation lifetime is linked to the model layer
count between 400 and 50 hPa, with higher layer counts being associated with longer perturbation lifetimes. Transport of
stratospheric water vapour emissions to the tropopause is a critical sink of the water vapour emissions, especially for the
models using prescribed tropospheric water vapour mixing ratios (GEOS-Chem, LMDZ-INCA, MOZART-3), where the
stratospheric water vapour tracer is effectively destroyed when it is transported into the model troposphere. We hypothesize
that the vertical model grid affects the modelling of the stratospheric to tropospheric transport, and furthermore that it
introduces a secondary sink which affects stratospheric water vapour. During the model integration the tropopause altitude
evolves over time, which causes parts of the model grid to switch from the stratosphere (evolving tracers) to troposphere
(prescribed ratios), stripping stratospheric tracers in the process. This has been noted to reduce water vapour perturbation
lifetimes of emissions near the tropopause in GEOS-Chem before (van 't Hoff et al. 2024a), and it also explains why we find
larger reductions (relative to the nominal scenario) in water vapour perturbation lifetimes in the low cruise scenario (S3) for
the models with coarser vertical grids

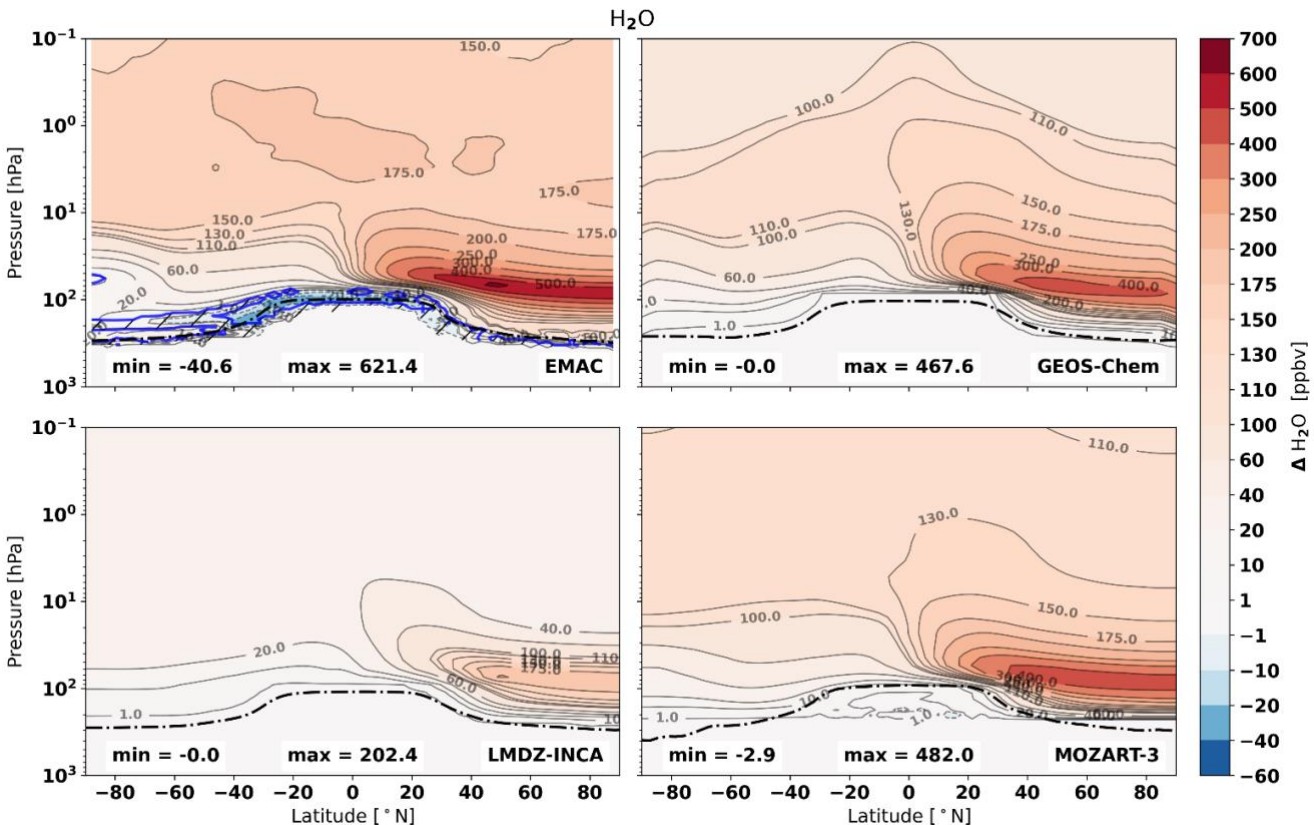

**Figure 2: Zonal mean changes in water vapour mixing ratios (ppbv) in response to the nominal supersonic emissions scenario (S1). Hatched areas enclosed by blue lines indicate regions that are not statistically significant for the EMAC results. The dash-dotted line indicates the mean tropopause pressure of each model. Similar figures for the triple NOₓ and low cruise scenarios are provided in the appendix (Figures A3 and A4).**

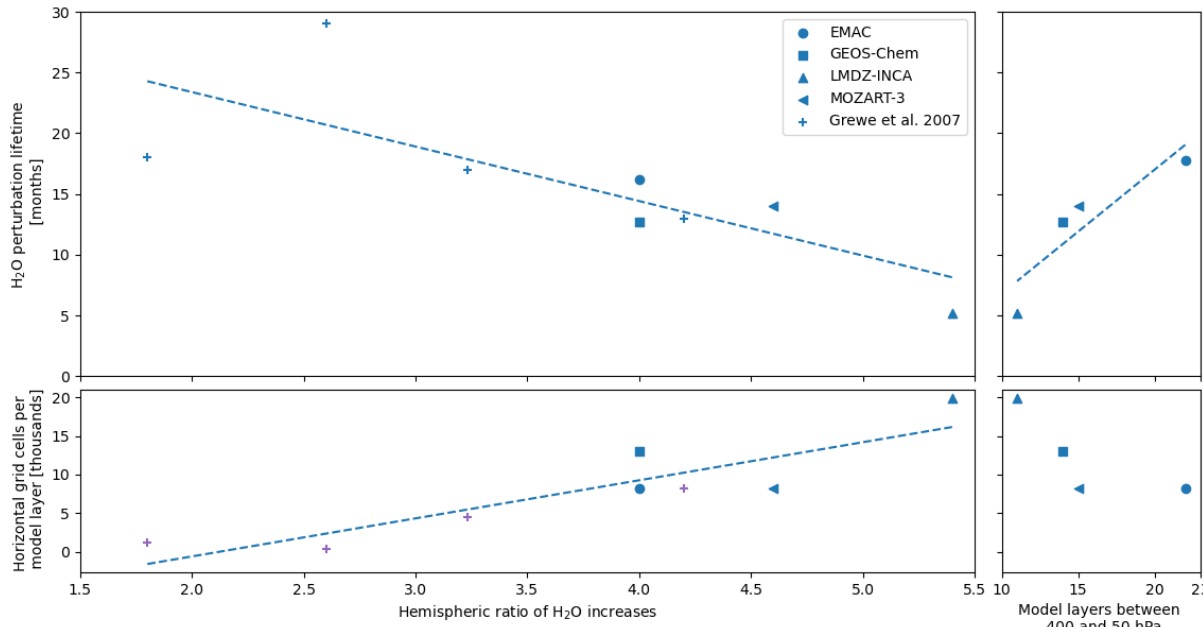

**Figure 3: Comparison of the H₂O perturbation lifetime in months and the hemispheric ratio of the H₂O increases of the nominal supersonic scenario (S1) with vertical and horizontal model grid characteristics. The top left figure shows the relationship between the perturbation lifetime and the hemispheric ratio, the top right the perturbation lifetime and the number of grid layers between 400 and 50 hPa, and the bottom left shows the hemispheric ratio and the horizontal grid fidelity. The bottom right figure shows the vertical layer count against the horizontal grid fidelity. Markers denote the different models. Results from Grewe et al. (2007) are included for their equivalent scenario (S5, Grewe et al. (2007)). The other emission scenarios are included in Figure A5.**

Figure 3 also shows that we find a trend between the hemispheric ratio and the horizontal grid fidelity. Strahan and Polansky (2006) found that the use of coarse horizontal grids led to overestimations of interhemispheric mixing in modelled atmospheres, and likewise we find smaller hemispheric ratios, suggesting that less water vapour emissions are transported to the southern hemisphere,e in the models with coarser horizontal grids. However, given the inverse relationship between the vertical and horizontal grid fidelities, we expect that this is primarily affected by the perturbation lifetime. The vast majority of water vapour emissions is in the northern hemisphere, therefore shorter perturbation lifetimes affect transport of the water vapour perturbation to the southern hemisphere, increasing the hemispheric ratio. We also see these trends in the responses to the other emissions scenarios (Figure A5), indicating that differences in model grids may be a significant contributor to differences in the lifetime and transport of high-altitude water vapour emissions.

### 4.3 Nitrogen oxides and reactive nitrogen

Figure 4 shows the perturbation of nitrogen oxides ($NO_x$) and reactive nitrogen ($NO_y = NO + NO_2 + NO_3 + HNO_2 + HNO_3 + HNO_4 + ClNO_3 + 2 N_2O_5 + PAN + ClNO_2 + BrNO_3$) from the nominal supersonic scenario (S1) over the four models. Similar figures for the triple $NO_x$ and low cruise scenarios are provided in the appendix (Figures A6 and A7). We find similar perturbations across all offline models. The $NO_x$ responses are primarily concentrated around the equator, with the strongest

accumulation in the middle-stratosphere and a secondary zone near the northern equatorial tropopause. In contrast to the $NO_x$ perturbation, the accumulation of $NO_y$ is concentrated around the region of cruise emissions. The $NO_y$ perturbation, which

includes that of $NO_x$, is mostly driven by increased formation of nitric acid ($HNO_3$) in these areas. $NO_y$ is then transported to the north pole or southwards to the tropical pipes (a region of upwelling over the tropics), where it makes its way to the middle stratosphere. This results in similar accumulation patterns for $NO_y$ as we find for the water vapour emissions.

Contrary to the offline models, EMAC predicts that the SST adoption leads to loss of $NO_x$ and $NO_y$ in the upper stratosphere

and southern hemisphere for all supersonic scenarios (Figures 4, A6, A7). We expect that these differences are predominantly driven by EMAC's use of online meteorology, which causes deviation of meteorological parameters between the baseline and the perturbed model run due to a combined result of the butterfly effect (noise) and meteorological feedbacks from the changes in stratospheric composition (Deckert et al. 2011). Figure 5 shows the differences in the EMAC temperature fields. It shows that EMAC's stratosphere cools in response to the three supersonic scenarios. The stratospheric cooling has several effects on

EMAC's chemistry, some of which are reflected in the $NO_x$ response. Near the south pole we see indications that the cooling facilitates increased formation of PSCs. We find regional depletion of gas phase $NO_y$ reservoirs associated with PSC chemistry ($ClONO_2$, $HNO_3$, $HNO_4$) and increases in liquid phase $HNO_3$ particles and solid phase particles like nitric acid trihydrate (NAT). These changes suggest that PSC chemistry is enhanced, increasing the sedimentation of stratospheric nitrogen compounds and leading to denitrification of the southern stratosphere. This likely drives the loss of $NO_x$ and $NO_y$ over the

south pole. Near the North pole similar responses may occur, but this is hard to discern due to the proximity of the emission sources. Above pressure altitudes of 10 hPa, where nudging is no longer applied, there is stratospheric cooling of over -0.3 K in response to the nominal SST emissions. The cooling may contribute to the loss of $NO_x$ and $NO_y$, as it slows down the N to $NO_x$ reformation reactions (Rosenfield and Douglass, 1998), but it likely also has more complex effects on the nitrogen chemistry cycles. Besides the change in temperature, there are also changes in EMAC's horizontal and vertical wind fields,

but since these are nudged they are predominantly statistically insignificant (Figures A8 to A10). Some changes in wind fields can be seen above 10 hPa, which may alter mixing in this region. Altogether, the use of online meteorology leads to a very different response of stratospheric $NO_x$ and $NO_y$ compared to the offline models. Given the sensitivity of ozone to $NO_x$, this is also linked to differences in the ozone response which we discuss next.

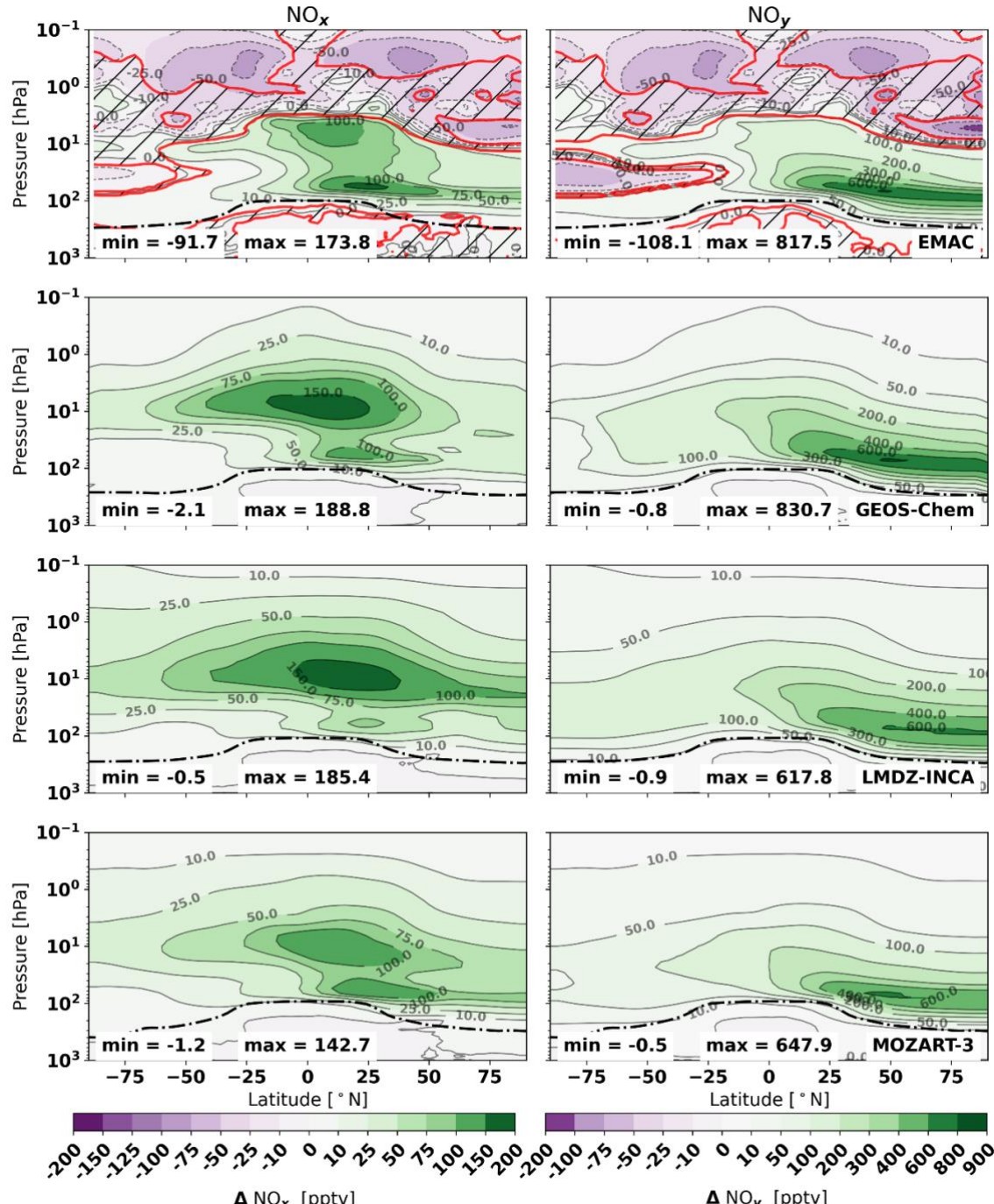

**Figure 4: Zonal mean changes in NOx (left) and NOy (right) mixing ratios (pptv) in response to the nominal supersonic emissions (S1). Top to bottom: EMAC, GEOS-Chem, LMDZ-INCA, MOZART-3. Hatched areas enclosed by red lines indicate regions that are not statistically significant for the EMAC results. The dash-dotted line indicates the mean tropopause pressure for each of the models. Figures for triple NOx (S2) and low cruise (S3) in the appendix (Figures A6, A7).**

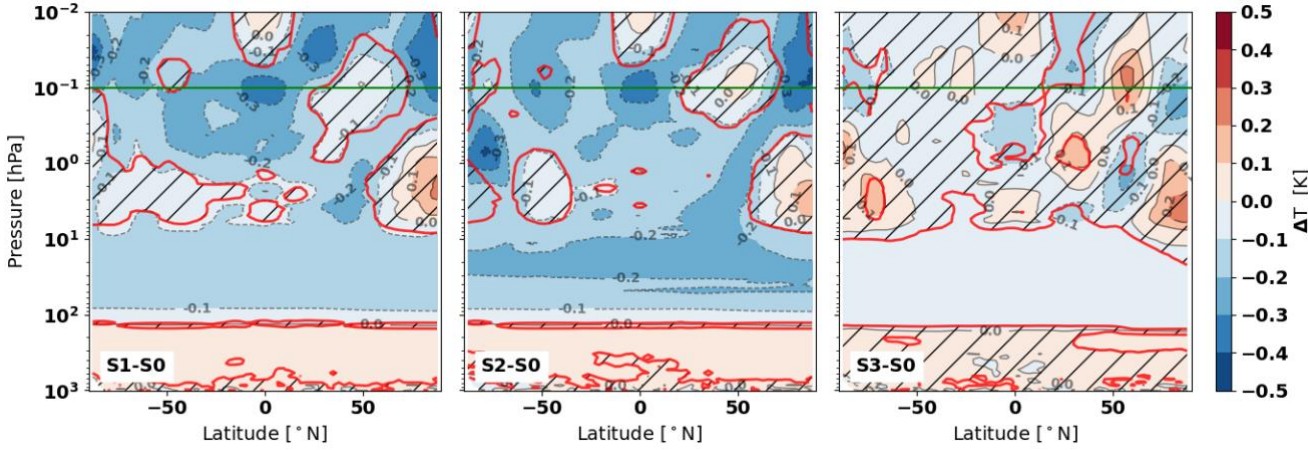


**Figure 5: Zonal mean changes in temperature (in Kelvin) in the EMAC model, in response to the nominal (left), triple NOₓ (middle), and low cruise (S3) scenarios. Hatched areas enclosed by red lines indicate regions that are not statistically significant. The green horizontal line at 0.1 hPa indicates the upper pressure shown in other figures.**

### 4.4 Ozone

Across all models and scenarios we find increases in lower-stratospheric ozone mixing ratios paired with ozone depletion in the upper-stratosphere (Figure 6). Similar patterns are found for the other supersonic scenarios (Figures 7, A11,A12). This pattern has been reported in several other studies, where the ozone increases in the lower-stratosphere are attributed to $NO_x$-driven ozone formation and the ozone layer's self-healing effect (Zhang et al. 2023; Eastham et al. 2022; Fritz et al. 2023; Zhang et al. 2021b). This increase is strongest in the LMDZ-INCA model where the ozone increase spans both hemispheres.

In GEOS-Chem and MOZART-3 this increase is limited to the equatorial lower-stratosphere underneath the main lobe of ozone depletion, and in EMAC it only occurs in the northern hemisphere.

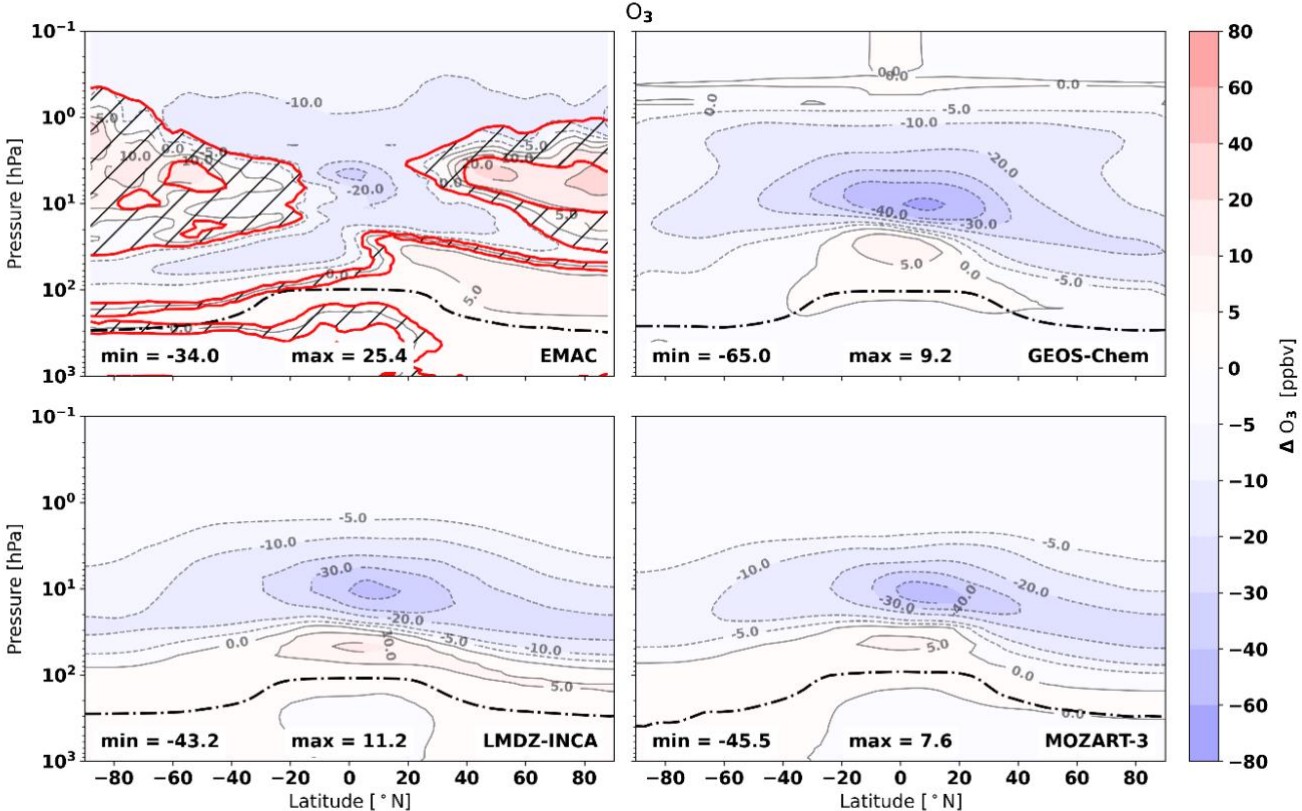

**Figure 6: Mean changes in ozone volume mixing ratio [ppbv] in response to the nominal supersonic (S1) emissions. Hatched areas enclosed by red lines indicate regions that are not statistically significant for the EMAC results. The dash-dotted line indicates the mean tropopause pressure for each of the models. Similar figures for the triple $NO_x$ and low cruise scenarios are provided in the appendix (Figures A11 and A12).**

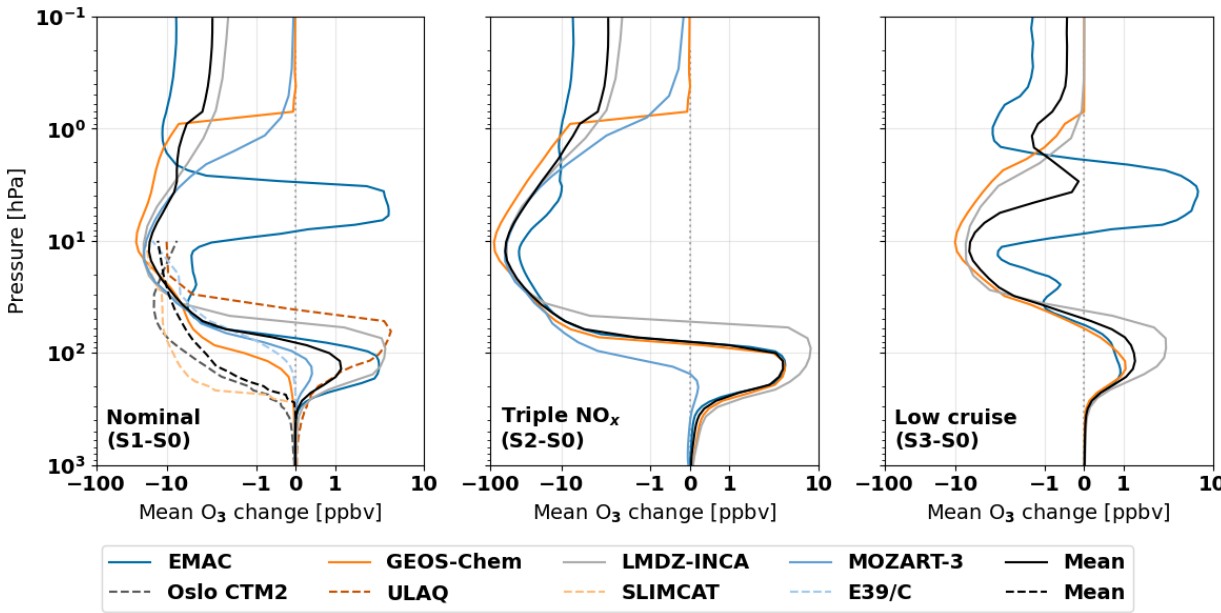

**Figure 7: Mean changes in ozone volume mixing ratio [ppbv] over altitude for the nominal supersonic (S1, left), triple NO$_x$ (S2, middle) and low cruise (S3, right) emission scenarios. Dashed lines show results from models used by Grewe et al. (2007).**

When the supersonic NO$_x$ emissions are tripled the effect on ozone is enhanced, particularly over the tropics in the middle-stratosphere (Figure A11). We find that there is a nonlinear relationship between the SST NO$_x$ emissions and the global ozone losses across all models. In GEOS-Chem and LMDZ-INCA the tripling of NO$_x$ emissions increases stratospheric ozone budget losses (Table A3) by factors of 2.4 and 2.6, respectively, whereas these factors are 3.8 and 7.4 for MOZART-3 and EMAC. Between the offline models MOZART-3 is most sensitive to NO$_x$ emissions, which could be related to its lower background

NO$_x$ levels. In the low cruise emissions scenario both the magnitude of ozone increases and losses are reduced (Figure A12).

Similar to the NO$_x$ and NO$_y$ responses, EMAC's ozone response differs from the offline models (Figs. 6, 7), which is likely coupled to feedbacks with its online meteorology. The most notable difference is the presence of ozone increases at high northern and southern latitudes above 10 hPa, both being partially statistically significant. We hypothesize that their formation

is driven by the previously discussed stratospheric cooling and enhancements of PSC chemistry. In the southern hemisphere, the meteorological feedbacks cause shifts in the abundance of nitrogen and chlorine (ClONO$_2$) reservoir species, which correlate with areas of ozone changes. In the northern lower stratosphere, the ozone increase correlates with an increase in HNO$_3$ formation. The patterns that should be associated with PSC chemistry tend to vanish for the triple NO$_x$ emission scenario (Figure A11), which may point towards their limited magnitude. The regions of ozone increases could also be related to

slowdowns of the Chapman mechanism due to cooling and perturbations in local transport (Kirner et al., 2014). The former is also seen in the odd O$_x$ loss rates, which are evaluated in the next section.

Differences between the model responses become more evident when the changes in ozone columns are compared. Figure 8 shows the mean ozone column change over latitude for the emission scenarios alongside the multi-model mean profile. It shows that the spread between the models is biggest in the northern hemisphere for all emission scenarios, particularly near the north pole. Expanding this into seasonal ozone column changes (Figure 9), we find that all models show different seasonal behaviour as well. For example, GEOS-Chem shows enhancement of ozone column loss during both the Arctic and Antarctic ozone hole formation. These enhancements are also present in LMDZ-INCA, albeit at a smaller scale. MOZART-3 does not show them, and it instead calculates the highest Arctic ozone depletion from June to November. EMAC shows year-round increases in the ozone column in the northern hemisphere. We expect that such differences are results of differences in the modelling of processes important to ozone, such as the PSC processes and feedbacks from emissions on heterogeneous chemistry. Only GEOS-Chem and LMDZ-INCA capture the effect of the emissions on PSC formation and the available surface area for heterogeneous chemistry, which may explain why only these models find enhancement of the seasonal ozone holes. The differences may further be affected by the availability of stratospheric halogens (Table A4). For example, the enhancement of the ozone hole is stronger in GEOS-Chem, which also has higher stratospheric halogen availability.

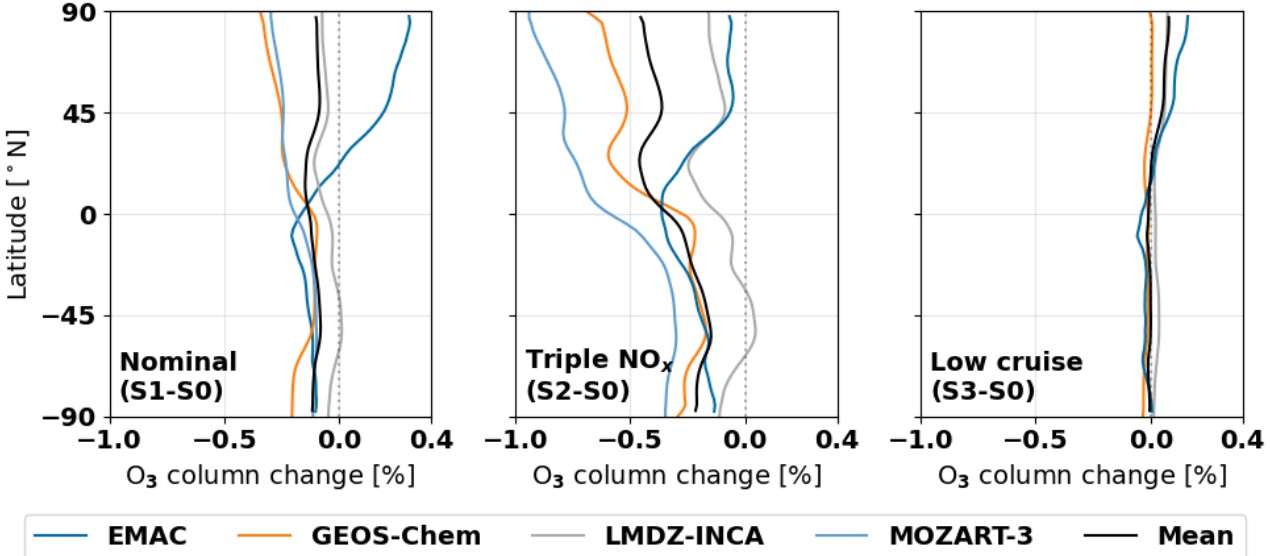

**Figure 8: Zonal mean annually averaged changes in ozone columns (percentage) for nominal supersonic (S1, left), triple NO$_x$ (S2, middle), and low cruise (S3, right) emission scenarios.**

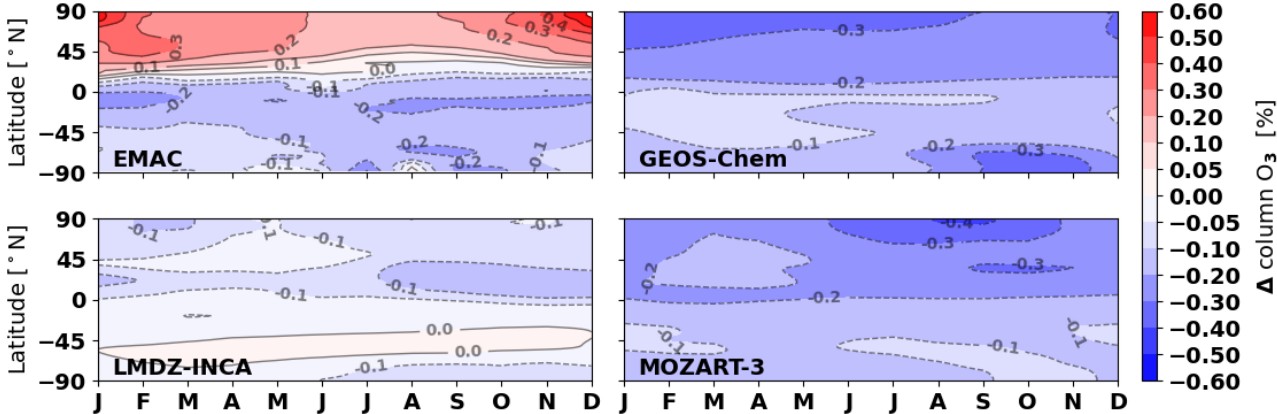

**Figure 9: Mean monthly changes in ozone columns (percentage) in response to the nominal supersonic emissions scenario (S1 – S0). Similar figures for the other emission scenarios are provided in the appendix (Figures A13 & A14)**

### 4.5 Odd oxygen loss

To better understand the source of the differences in the ozone responses we evaluate the changes in odd oxygen ($O_x$) reaction rates for EMAC, GEOS-Chem, and LMDZ-INCA. We use the same reaction grouping as Zhang et al. (2023; 2021a). These

models all find that the supersonic scenarios lead to net-increases in $O_x$ loss. When averaged over altitude, we find similar baseline $O_x$ loss rates across all models (Figure 10), and the responses of the GEOS-Chem and LMDZ-INCA models appear to share similar profiles whereas that of EMAC is very different. When the $O_x$ loss responses are seen as zonal averages (Figure 11), further differences in the spatial distribution of the $O_x$ loss responses become evident.

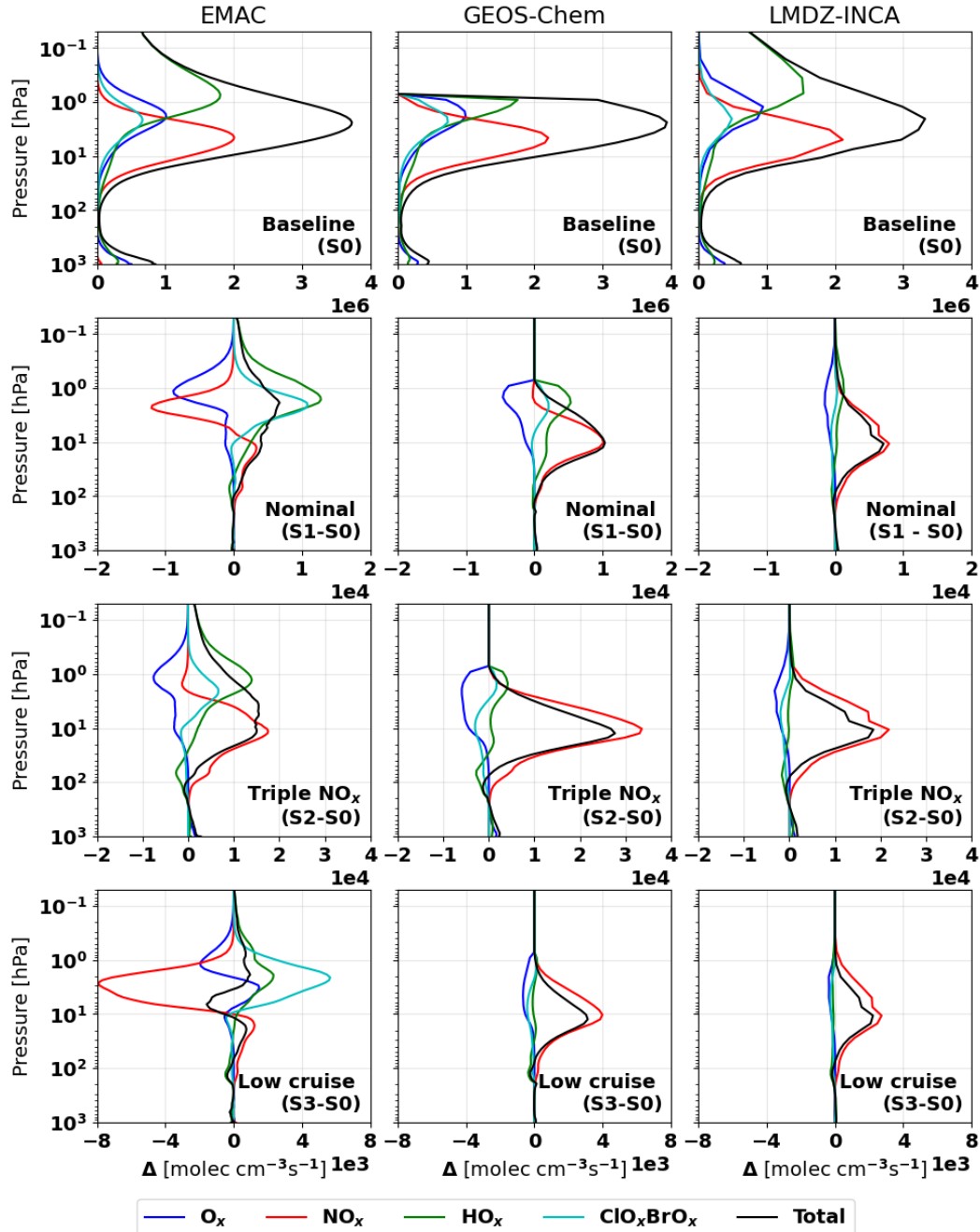

**Figure 10: Mean background $O_x$ loss rates (top) and $O_x$ loss rate perturbations from emission scenarios over pressure altitude for EMAC (left), GEOS-Chem (middle) and LMDZ-INCA (right). From top to bottom: background loss rates, loss rate perturbation from nominal supersonic emissions (S1), loss rate perturbation from triple $NO_x$ emissions (S2), loss rate perturbation from low cruise emissions (S3).**


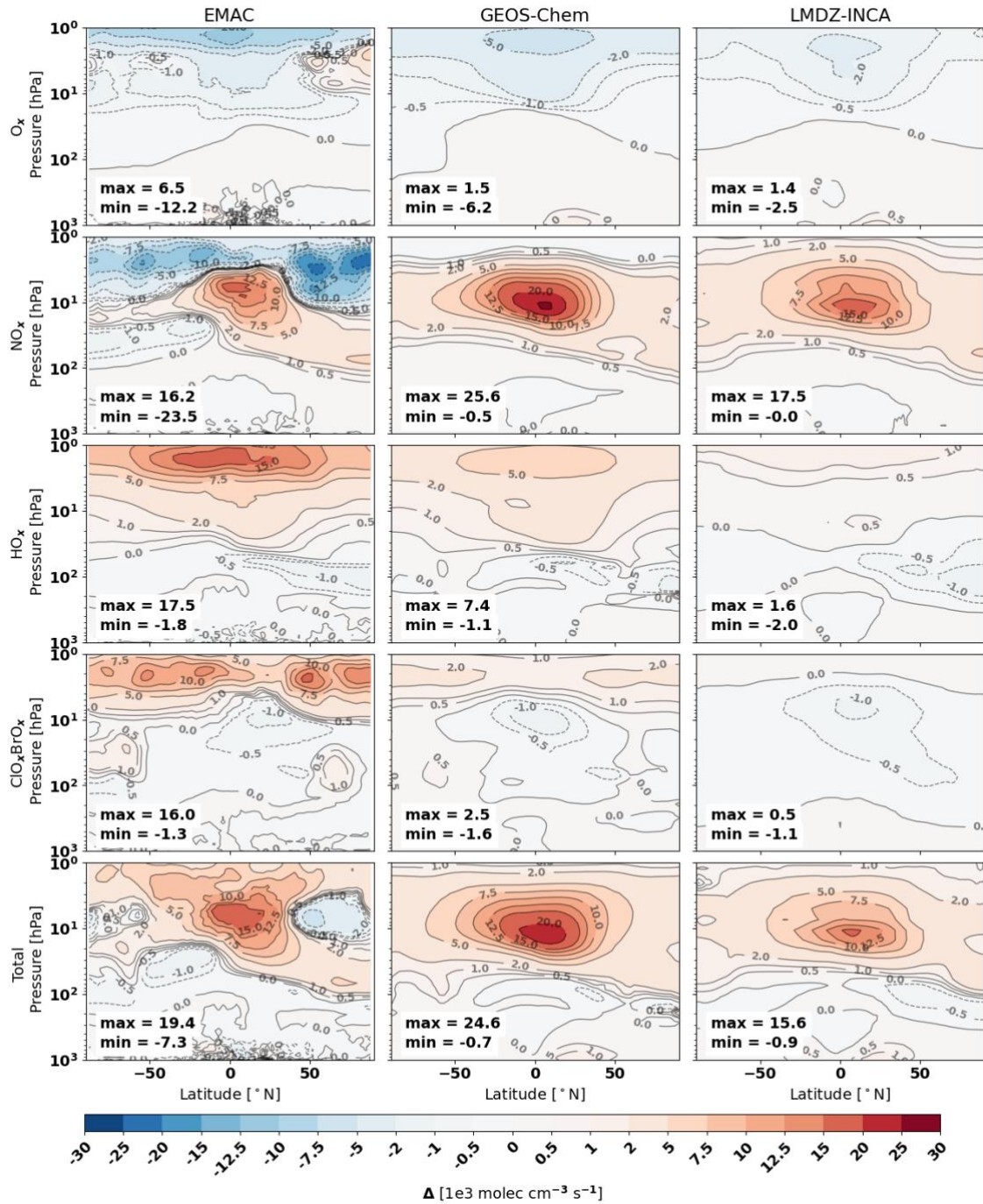

**Figure 11: Zonal average mean changes in Ox loss reactions in response to the nominal supersonic emissions for EMAC (left), GEOS-Chem (middle) and LMDZ-INCA (Right). Loss perturbations are split into $O_x$-driven loss (top), $NO_x$-driven loss (second row), $HO_x$-driven loss (third), $ClO_xBrO_x$ driven loss (fourth) and total changes in $O_x$ loss (bottom). Values in molec cm$^{-3}$ s$^{-1}$. Similar figures for the triple NOx (S2) and low cruise (S3) scenarios are provided in the appendix (Figures A15 & A16).**

The $O_x$ loss reaction responses of the GEOS-Chem and LMDZ-INCA models are similar. In these models, the largest response to the nominal scenario is an increase in $NO_x$-driven $O_x$ losses from 200 to 20 hPa. $HO_x$-driven losses also increase, but mostly at higher altitudes from 20 hPa to 0.1 hPa. These increases are paired with decreases in $O_x$-$O_x$ losses, as the availability of $O_x$ reduces. The models also calculate increases in $ClO_xBrO_x$ driven losses above 10 hPa. In response to the triple $NO_x$ scenario, the $NO_x$-driven $O_x$ losses increase by around threefold, reducing the effect on $ClO_xBrO_x$-driven losses. For the lower cruise altitude scenario (S3) we find similar changes in $O_x$ loss rates as the nominal supersonic scenario but at smaller magnitudes. In the case of the GEOS-Chem and LDMZ-INCA models, the reduction in cruise altitude also sharply reduces $HO_x$-driven $O_x$ losses. This is related to the shorter water vapour perturbation lifetimes at this cruise altitude.

Figures 10 and 11 show that the increased complexity of EMAC's response extends to odd $O_x$ loss rates. Contrary to the previously shown changes in ozone and $NO_x$ mixing ratios, the response in EMAC's $O_x$ loss reactions is entirely statistically significant. Both figures show that there are areas where $NO_x$-driven $O_x$ losses are reduced, coinciding with the previously discussed areas of denitrification in the upper-stratosphere and over the south pole (Figure 4). The reduction in $NO_x$-driven $O_x$ losses is coupled with local increases of $HO_x$ and $ClO_xBrO_x$ driven losses, the latter of which is also affected by the enhancement of PSC chemistry. The cooling of the stratosphere also slows the Chapman mechanism, leading to reductions in upper-stratospheric $O_x$-$O_x$ losses, with the exceptions of the areas where there is a net increase of ozone and therefore also $O_x$ availability.

The perturbation of $ClO_xBrO_x$-driven $O_x$ losses differs across the three models, with EMAC's $ClO_xBrO_x$ response showing peak values up to five times larger than GEOS-Chem, whereas $ClO_xBrO_x$ losses are mostly unaffected in LMDZ-INCA. The large response in EMAC is likely coupled to the local decreases in $NO_x$-driven losses in these areas, which are of similar magnitude (Figure 11). Between GEOS-Chem and LMDZ-INCA, we expect that these differences may be related to the availability and distribution of halogens. Table A4 (appendix) shows mean background mixing ratios for key halogens at the surface and from 200 to 10 hPa in the models' baseline atmospheres. While we find similar background halogen levels near the surface, GEOS-Chem has lower stratospheric CFC mixing ratios, and higher values of other halogens compared to LMDZ-INCA. This could suggest that CFC destruction is faster GEOS-Chem, which may affect the role of $ClO_xBrO_x$-driven $O_x$ losses. Another difference is the upper limit of the $O_x$ chemistry domain, which is around 1 hPa in GEOS-Chem but extends further in the other models . Above this altitude GEOS-Chem has no $O_x$ chemistry, yet other species related to ozone chemistry are allowed to evolve freely. This may lead to an accumulation of $HO_x$ and halogens in the mesosphere, contributing to increased $HO_x$ and $ClO_xBrO_x$ driven $O_x$ losses when they are transported downwards into the region of $O_x$ chemistry.

### 4.6 Radiative Forcing

From non-$CO_2$ emissions we estimate a net-warming effect of 13.9 mW/m$^2$ for the nominal supersonic scenario (S1). This is predominantly driven by the accumulation of stratospheric water vapour (20.8 mW/m$^2$) and warming from changes in the

distribution of ozone (3.2 mW/m$^2$). The calculated ozone forcing ranges from 1.3 mW/m$^2$ (GEOS-Chem) to 6.8 mW/m$^2$ (LMDZ-INCA), where EMAC and MOZART-3 find forcings of 3.0 and 2.2 mW/m$^2$ respectively. The largest spread is in the water vapour forcing, ranging from 6.2 mW/m$^2$ (LMDZ-INCA) to 32.4 mW/m$^2$ (MOZART-3). This is affected by the

differences in water vapour perturbation lifetime within the models, but also by the radiative schemes used to assess RF. For example, the two radiative schemes applied to EMAC find water vapour forcings of 29.7 and 22.2 mW/m$^2$ in this scenario, a relative difference of 33%. The SOCRATES model used with MOZART-3 also finds a larger water vapour forcing relative to the water vapour perturbation and lifetime. This suggests that the differences between radiative schemes may be a more important contributor to uncertainties in radiative forcing assessments than differences in atmospheric perturbations calculated

by CTMs or CCMs.

Only the GEOS-Chem and LMDZ-INCA models assess the effect of aerosols on RF, and the aerosol perturbations in these models are shown in Figures A17 and A18. These models do assess aerosol RF at different altitudes, which likely causes them to find differences forcing from BC aerosols (Eastham et al. 2022; Speth et al. 2021). For the nominal scenario LMDZ-INCA

calculates RF of 0.5 mW/m$^2$ from BC and -10.0 mW/m$^2$ from inorganic aerosols, and GEOS-Chem calculates -1.3 mW/m$^2$ and -9.3 mW/m$^2$ respectively. Between the models we calculate a mean aerosol forcing of -10.1 mW/m$^2$, resulting in a net forcing of 13.9 mW/m$^2$ from the nominal supersonic scenario. Considering the altitude dependency of the BC forcing, this value may change by up to ±0.5 mW/m$^2$ depending on the assessment altitude.

The tripling of NO$_x$ emissions (S2) has affects the radiative effect of ozone, which increases to a model-mean of 11.4 mW/m$^2$ (range 6.9 to 20.9). In this case we calculate model-mean RF from water vapour of 18.2 mW/m$^2$ (model range 6.3 to 31.1) and RF from aerosols of -0.4 mW/m$^2$ for BC and -9.7 mW/m$^2$ for inorganic aerosols. This results in a model-mean net RF of 19.4 mW/m$^2$ from non-CO$_2$ emissions (model range 16.7 to 28.1). The reduction of the cruise altitude and speed (S3) reduces the RF from ozone to 2.9 mW/m$^2$ (model range 2.1 to 4.6) and water vapour to 4.3 mW/m$^2$ (model range 0.7 to 8.2). RF from

aerosols is also smaller, with -0.1 mW/m$^2$ from BC and -3.3 mW/m$^2$ from inorganic aerosols. This results in a model-mean net RF of 3.8 mW/m$^2$ from non-CO$_2$ emissions (range 0.5 to 7.1).

**5 Discussion**

Across all models and assessments, we find that the partial replacement of subsonic aviation with supersonic aircraft leads to extensive changes to the atmospheric composition, particularly in the stratosphere, and global radiation budgets. The extent of

these effects appears to scale with fleetwide NO$_x$ emissions and the cruise altitude, increases of which enhance ozone column loss and associated radiative effects in all models. Therefore, we also find the largest effect on the ozone column and RF in response to the Mach 2 concept with higher NO$_x$ emissions (Scenario S2). In terms of NO$_x$ emissions, this scenario is closest

to SST concepts studied in other recent works (Zhang et al. 2023, 2021a; Eastham et al. 2022; Speth et al. 2021), which is why we consider it as a basis for comparison with literature and as the most plausible outlook for future SST adoption.

The stratospheric changes we identify in response to this scenario match patterns identified in several recent works (van 't Hoff 2024a; Zhang et al. 2023, 2021a; Eastham et al 2022; Kinnison et al. 2020). Here, we find a model-mean change in the global ozone column of -0.3% (-0.9 DU). Scaling by fuel consumption, this is similar to the -0.74% ozone column loss reported by Zhang et al (2023), who considered a larger SST fleet with around 2.1 times the fuel burn. It does not match with results from Eastham et al. (2022), who reported a larger ozone column loss (-0.77%) for a smaller SST fleet (14.9 Tg of annual SST fuel burn). This may be related to differences in background conditions and in the emission scenarios. The scenarios considered by Eastham et al. (2022) have higher SST $NO_x$ emissions, and they consider a more prominent role of the Asian market in SST adoption than the inventories we use, displacing more SST traffic to lower latitudes (Speth et al. 2021). The ozone column is substantially more sensitive to SST emissions near the tropics (van 't Hoff 2024a; Fritz et al. 2022), which may explain why they find higher ozone loss relative to the fuel consumption. The nominal and triple $NO_x$ scenarios we evaluate are also similar to scenarios evaluated by Zhang et al. (2021b, case A & C). In comparison to their results, we find also find similar ozone column losses (-0.1 % and -0.3%, compared to their -0.2% and -0.4%). We also find similar $O_x$ loss rate perturbations, in particular between GEOS-Chem and the WACCM4 model they used. Several recent studies have identified that the perturbation of ozone can be the primary source of radiative forcing from SST emissions (van 't Hoff 2024a; Zhang et al. 2023; Eastham et al. 2022), but we instead find water vapour to be dominant in all scenarios, matching the results from Zhang et al. (2021b) and Grewe et al. (2007). This is not agreed upon in all models however, as in some cases LMDZ-INCA and GEOS-Chem find that the ozone perturbation is the primary forcer (Scenarios S1 to S3 for LMDZ-INCA, scenario S3 for GEOS-Chem). We expect that this difference is related to fleetwide $NO_x$ emissions. Previous works have shown that RF from ozone perturbations scales with fleetwide $NO_x$ emissions (van 't Hoff 2024a; Zhang et al. 2021b), and even in in the triple $NO_x$ scenario our $NO_x$ emissions index is lower than indices used by the works that find forcing from ozone to be dominant (van 't Hoff 2024a; Zhang et al. 2023; Eastham et al. 2022). Therefore, it is plausible that we'd also find the primary radiative effect to be from ozone if higher supersonic $NO_x$ emissions are considered.

Since the nominal scenario that we consider is almost identical to scenario S5 from Grewe et al. (2007) we also compare to their results, although we note that there are considerable differences between our models and the ones they used. For example, our models have higher resolutions (horizontal and vertical) and upper grid levels (0.1 to 0.01 hPa, compared to their 10 hPa). Compared to their results, we find lower stratospheric perturbations of water vapour (47 Tg compared to 64 Tg) and ozone (-3.1 Tg compared to -8 Tg). Considering the perturbation mass and the hemispheric ratios, we also find smaller spread in our models compared to theirs (Figure A19). In terms of radiative forcing we calculate this at the tropopause level, whereas they calculated it at the top of the atmosphere, hindering direct comparison. For the scattering inorganic aerosols, which should not be affected much by the assessment altitude, we find similar RF (-9.7 mW/m$^2$ compared to their -11.4 mW/m$^2$). At the top of

the atmosphere we find smaller RF for black carbon (0.5 mW/m$^2$ for LMDZ-INCA, 1.7 mW/m$^2$ for GEOS-Chem, compared to their 4.8 mW/m$^2$), which could be related to differences in radiative modelling, aerosol size distributions, or the simulated transport of the black carbon.


Like earlier works, we find that the perturbation of stratospheric water vapour plays a key role in the radiative effect of SSTs (Zhang et al. 2023; Eastham et al. 2022; Matthes et al. 2022; Grewe et al. 2007), but we note that RF from water vapour is prone to several uncertainties. Foremost, we find that the RF depends on the radiative schemes, with relative differences of up to 30% between the two radiative schemes which we apply to EMAC. The radiative effect from water vapour is also directly

related to the stratospheric water vapour burden, and therefore it depends on the water vapour perturbation lifetime within the models. We find indications that this lifetime is affected by the vertical resolution and model grid, linking increased grid layers between 400 and 50 hPa to higher perturbation lifetimes. Therefore, our results suggest that both the model grid itself and the choice of radiative scheme may be important contributors to uncertainties surrounding radiative effects of water vapour emissions. We expect that these factors may be influential in all assessments of high-altitude water vapour emissions, and not

exclusively to supersonic aircraft.

We find considerable differences between the responses of the online and offline models. Between the offline models (GEOS-Chem, LMDZ-INCA, MOZART-3) we find good agreement in all perturbations of the stratospheric composition. The most notable difference is the lower vertical domain of GEOS-Chem's extensive stratospheric ozone chemistry, which may lead to

increased influx of HO$_x$ at the upper-stratospheric boundary, although our results do not indicate that this has a large effect on the calculated ozone column responses. Comparing the online EMAC model to the offline models, we see some substantial differences from the inclusion of meteorological feedbacks (predominantly stratospheric cooling) on chemistry. The inclusion of this feedback allows EMAC to capture interactions that are not included in the offline models. For example, we find that the stratospheric cooling enhances PSC chemistry in EMAC, leading to denitrification over the south pole in response to the

SST adoption. To our knowledge, this feedback from SST emissions has not previously been identified in earlier works. We also identify denitrification of the upper-stratosphere in response to the SST emissions, likely due to slowdown of NO$_x$ reformation reactions and interactions with nitrogen chemistry cycles. This contributes to increases in upper-stratospheric ozone at high latitudes, which has also been shown other works that use EMAC in a similar configuration (Pletzer et al. 2022; Kirner et al. 2014). We find these differences even when the meteorological feedbacks are still constrained. Within the model

the horizontal and vertical winds are still nudged for the majority of the stratosphere, and furthermore some feedbacks like local temperature changes from black carbon perturbations are not included. The inclusion of these feedbacks would likely further alter the response to high-altitude emissions. We expect that the consideration meteorological feedbacks might be critical to the complete assessment of the effects of high altitude emissions, as some important feedbacks may be overlooked otherwise.


The adoption of a fleet of SSTs should be considered in context of other options for air travel, both in terms of their $CO_2$ and non-$CO_2$ effects. For the nominal and triple $NO_x$ scenarios, we calculate a fuel burn to RPK ratio of 79.3 g/ RPK, and for the low cruise concept a ratio of 58.0 g/ RPK. In comparison, from Lee et al. (2021) we calculate a fuel burn to RPK ratio of around 38.0 g/ RPK for the 2018 subsonic fleet. This suggests that replacing a RPK with these SST concepts increases the associated fuel consumption, and by extension the emission of $CO_2$ and its radiative effects, by 109% and 53% for the respective nominal and low cruise concepts. A similar trend also holds for the radiative effects of non-$CO_2$ emissions. In the triple-$NO_x$ scenario we find that replacing $7.3 \times 10^{11}$ subsonic RPK with SSTs increases RF from non-$CO_2$ emissions by 19.4 $mW/m^2$. From this we calculate an increase in the RF:RPK ratio of $26.6 \times 10^{-12}$ $mW/m^2$ / RPK. In case of the nominal and low cruise scenarios we find increases in this ratio of $19.0 \times 10^{-12}$ and $5.2 \times 10^{-12}$ mW /$m^2$ / RPK, respectively. These estimates incorporate the removal of the equivalent RPKs from the subsonic fleet, thereby representing the additional RF from non-$CO_2$ emissions per RPK when an RPK is flown by a SST rather than a subsonic aircraft. We note that our results do not reflect the RF benefits of practically eliminating contrail impacts from the subsonic RPK, but do reflect aerosol RF. In comparison, using non-$CO_2$ RF estimates from the 2018 subsonic aviation from Lee et al. (2021), we calculate a RF:RPK ratio of $14.4 \times 10^{-12}$ $mW/m^2$ / RPK for subsonic radiative effects from non-$CO_2$ emissions. Our results therefore indicate that the replacement of subsonic RPKs with the triple $NO_x$ scenario SST would increase the non-$CO_2$ RF:RPK ratio by 185% compared to the estimate of Lee et al. (2021), and that the nominal SST and low cruise (Mach 1.6) SST would increase the RPK cost by 132% and 36%, respectively. These discrepancies would differ if contrails were to be included in the RF assessment, but they provide an estimate of the additional climate impacts of SSTs over subsonic aircraft. The disparity that we identify has also been reported in earlier works (Eastham et al. 2022; Speth et al. 2021; Grewe et al. 2007), and while our results suggest that this disparity may be mitigated by reducing supersonic $NO_x$ emissions, cruise speed, and altitude, we expect it will nonetheless persist due to the more sensitive emission altitudes, the higher fuel requirements, and the lower passenger numbers of supersonic aircraft.

Our results provide some actionable information for consideration in sustainability discussions related to SSTs. We find the atmospheric and radiative effects are predominantly driven by $NO_x$ and water vapour emissions, which indicates that the use of sustainable aviation fuels is not likely to lead to substantial differences in these effects. On the contrary, sustainable aviation fuels are likely to have lower sulphur and black carbon emissions, which will increase SST radiative effects by reducing the emissions responsible for the cooling RF, as also identified by Speth et al. (2021). We also remark that the effect on the ozone column could be considered in the context of the effects on human health. For example, in response to the triple-$NO_x$ scenario, we find a model-mean global ozone column loss of -0.3 % (-0.9 DU), but some models calculate year-round depletion of up to -0.7% (-2.1 DU) over the northern hemisphere. Considering the distribution of population, the effect on human health is likely larger than what the global average would imply. Estimating the effect on human health lies outside of the scope of this work, but it may be an effective means to communicate the effect of changes in the ozone column. Such an approach may also

account for the changes in air quality from tropospheric ozone perturbations, which are otherwise not included in discussions surrounding global column ozone perturbations.

## 6 Conclusions

For the first time since 2007, we present a comprehensive multi-model assessment of the effects of the partial replacement of subsonic aviation traffic with a fleet of supersonic transport aircraft on atmospheric composition and global radiative forcing. With four widely-used models (EMAC, LMDZ-INCA, GEOS-Chem, and MOZART-3) we evaluate three supersonic adoption scenarios based on the emission scenarios of the SCENIC project (Grewe et al. 2007). Two of these scenarios consider the adoption of a Mach 2 supersonic aircraft operating at cruise altitudes of 16.5 to 19.5 km to replace around 4% of subsonic aviation traffic, differing in fleetwide $NO_x$ emissions (13.80 and 4.60 g($NO_2$) /kg). The third scenario considers aircraft with a lower cruise speed (Mach 1.6) and altitude instead (13.1 to 16.7 km).

The partial replacement of subsonic aviation with both Mach 2 concepts results in a reduction in the global ozone column. For the Mach 2 concept with $NO_x$ emissions of 13.80 g ($NO_2$)/kg we calculate model-mean global ozone column loss of -0.3% (-0.9 DU), with higher losses across the northern hemisphere (model-mean up to -0.5%, -1.5DU). The replacement of subsonic aviation with this concept increases radiative forcing by of 19.4 mW/m². The biggest forcing is from changes in stratospheric water vapour (18.2 mW/m²), followed by ozone (11.4 mW/m²), and aerosols (-10.2 mW/m²). If the fleetwide $NO_x$ emissions are reduced by 67%, the net forcing also reduces to 13.9 mW/m² because of the smaller ozone perturbation (-0.1% (-0.3 DU)) and its associated forcing (3.2 mW/m²). If part of the subsonic aviation is instead replaced by the Mach 1.6 concept, which has a lower cruise altitude and fleetwide $NO_x$ emissions, the effects on stratospheric composition and radiative forcing are reduced by 3.8 mW/m². These values do not account for potential changes in contrail formation and the increase in $CO_2$ emissions. Compared to estimates of subsonic aviation, we find that the replacement of subsonic passenger revenue kilometres with supersonic aircraft increases the associated radiative forcing from non-$CO_2$ emissions by up to 185% compared to subsonic aircraft.

Compared to the previous multi-model assessment of the atmospheric and radiative effects of supersonic aircraft (Grewe et al. 2007), we see a smaller spread in our model evaluations of the water vapour and ozone perturbations. We find good agreement in the composition changes between the three models which use offline meteorology, but we also see large differences with the model with online meteorology (EMAC). The inclusion of meteorological feedbacks in the model captures several responses to the emissions that are not captured in the offline models, leading to denitrification of the upper-stratosphere and south pole. These feedbacks have substantial effects on the stratospheric ozone and nitrogen responses, and we expect that they may be of critical importance to the assessment of the effects of high-altitude emissions.

## 7 Acknowledgements

This work was funded by the European Union's Horizon 2020 research and innovation programme. Authors (J.A.H, R.T., J.P., V.G., D.H., I.C.D., M.M.M.) were funded by the MORE & LESS project (MDO and REgulations for Low-boom Environmentally Sustainable Supersonic aviation, grant No. 101006856). Others (A.S. & S.M.) were funded by the SENECA project ((LTO) Noise and Emissions of Supersonic Aircraft, grant No.101006742). The GEOS-Chem simulations were supported by the Dutch national e-infrastructure and supercomputer with the support of the SURF Cooperative (Grant no. EINF-5945). The EMAC model simulations were computed at the German Climate Computing Center (DKRZ). The resources for the simulations were provided by the German Bundesministerium für Bildung und Forschung (BMBF). Ruben Rodriguez de Leon is thanked for maintaining and running the SOCRATES RTM.

## 8 Data availability

The data supporting the results of this work is publicly available at DOI: 10.4121/dd38833d-6c5d-47d8-bb10-7535ce1eecf1 (van 't Hoff et al. 2024b). Reserved DOI for the dataset to be minted on acceptance. Reviewers can preview the dataset through https://data.4tu.nl/private_datasets/_qffTJdRitRConukkBmBTegk9CJTJZD36xRUN6lif5M.

## 9 Author contribution

Conceptualization: ICD, VG, DH, JAH, RNT; Formal analysis: DH, JAH, JP, AS; Investigation: ICD, VG, DH, JAH, SM, JP, AS; Data Curation: DH, JAH, MMM, JP, AS, RNT; Writing - Original Draft: JAH; Writing - Review & Editing: all; Visualization: JAH, JP; Supervision: ICD; Funding acquisition: ICD, VG, DH, SM, AS.

## 10 Competing interests

The authors declare that they have no conflict of interest.

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

## Appendices

**Table A1: Summary of stratospheric $H_2O$ perturbations in all scenarios. Values calculated as triannual averages**

| | Background $H_2O$ | $H_2O$ perturbation | Perturbation lifetime | Hemispheric ratio | $H_2O$ Increase Hemispheric ratio |
|---|---|---|---|---|---|
| **Nominal (S1)** | [Tg] | [Tg] | [months] | [NH/SH] | [NH/SH] |
| EMAC | 4133.5 | +63.3 (+1.5 %) | 16.2 | 5.9 | 4.0 |
| GEOS-Chem | 7344.8 | +49.3 (+0.7 %) | 12.7 | 4.0 | 4.0 |
| LMDZ-INCA | 3743.8 | +20.1 (+0.6 %) | 5.2 | 5.4 | 5.4 |
| MOZART-3 | 3519.8 | +54.7 (+1.6 %) | 14.0 | 4.6 | 4.6 |
| Model-mean | | +46.9 (+1.1 %) | 12.0 | 5.0 | 4.5 |
| **Triple NO$_x$ (S2)** | | | | | |
| EMAC | 4133.5 | +61.8 (+1.5 %) | 15.8 | 6.6 | 4.1 |
| GEOS-Chem | 7344.8 | +49.8 (+0.7 %) | 12.8 | 3.9 | 3.9 |
| LMDZ-INCA | 3743.8 | +20.6 (+0.6 %) | 5.3 | 5.2 | 5.2 |
| Model-mean | | +44.1 (+0.9 %) | 11.3 | 5.2 | 4.4 |
| **Low cruise (S3)** | | | | | |
| EMAC | 4133.5 | +16.0 (+0.4 %) | 9.1 | 3.2 | 3.1 |
| GEOS-Chem | 7344.8 | +6.0 (+0.1 %) | 3.4 | 10.3 | 10.4 |
| LMDZ-INCA | 3743.8 | +2.4 (+0.1 %) | 1.3 | 36.4 | 38.1 |
| Model-mean | | +8.1 (+0.2 %) | 4.6 | 16.6 | 17.20 |

**Table A2: Summary of stratospheric NOₓ perturbations due to the emission scenarios. Values calculated as triannual averages.**

| | Background $NO_x$ | $NO_x$ perturbation | Perturbation lifetime | $NO_x$ increase hemispheric ratio |
|---|---|---|---|---|
| **Nominal (S1)** | [Gg $NO_2$] | [Gg $NO_2$] | [months] | [NH/SH] |
| EMAC | 2329.6 | +37.4 (+1.6 %) | 4.0 | 4.5 |
| GEOS-Chem | 2302.0 | +43.5 (+1.9 %) | 4.7 | 2.7 |
| LMDZ-INCA | 2457.8 | +42.6 (+1.7 %) | 4.6 | 2.3 |
| MOZART-3 | 2035.7 | +32.1 (+1.6 %) | 3.5 | 6.7 |
| Model-mean | | +38.9 (+1.7%) | 4.2 | 4.1 |
| **Triple $NO_x$ (S2)** | | | | |
| EMAC | 2329.6 | +140.5 (+6.0 %) | 3.3 | 3.7 |
| GEOS-Chem | 2302.0 | +173.6 (+7.5 %) | 4.1 | 2.5 |
| LMDZ-INCA | 2457.8 | +119.5 (+4.9 %) | 2.8 | 2.3 |
| MOZART-3 | 2035.7 | +112.8 (+5.5 %) | 2.6 | 6.2 |
| Model-mean | | +136.6 (+6.0 %) | 3.2 | 3.7 |
| **Low cruise (S3)** | | | | |
| EMAC | 2329.6 | +18.2 (+0.8 %) | 4.6 | 3.7 |
| GEOS-Chem | 2302.0 | +23.7 (+1.0 %) | 6.0 | 2.7 |
| LMDZ-INCA | 2457.8 | +17.4 (+0.7 %) | 4.4 | 2.3 |
| Model-mean | | +19.8 (+0.8 %) | 5.0 | 2.9 |

**Table A3: Summary of O₃ perturbations due to the emission scenarios. Values calculated as triannual averages.**

| | Strat. background O3 | Strat. $O_3$ perturbation | $O_3$ increase hemispheric mass ratio | $O_3$ loss hemispheric ratio | Background column | Column perturbation | Column $O_3$ loss hemispheric ratio |
|---|---|---|---|---|---|---|---|
| **Nominal (S1)** | [Tg] | [Tg] | [NH/SH] | [NH/SH] | [DU] | [DU] | [NH/SH] |
| EMAC | 3137.3 | -1.1 (-0.0 %) | 4.4 | 0.8 | 340.0 | +0.0 (+0.0 %) | 0.9 |
| GEOS-Chem | 3002.9 | -5.9 (-0.2 %) | 1.3 | 2.0 | 321.9 | -0.7 (-0.2 %) | 2.4 |
| LMDZ-INCA | 3092.0 | -1.8 (-0.1 %) | 1.3 | 1.8 | 328.5 | -0.2 (-0.0 %) | 2.6 |
| MOZART-3 | 2926.5 | -3.7 (-0.1 %) | 0.4 | 2.6 | 331.2 | -0.6 (-0.2%) | 19.4 |
| Model-mean | | -3.1 (-0.1 %) | 1.9 | 1.8 | 330.4 | -0.3 (-0.1 %) | 6.3 |
| **Triple NO$_x$ (S2)** | | | | | | | |
| EMAC | 3137.3 | -8.1 (-0.3 %) | 5.4 | 1.5 | 340.0 | -0.6 (-0.2 %) | 1.3 |
| GEOS-Chem | 3002.9 | -14.1 (-0.5 %) | 1.1 | 1.9 | 321.9 | -1.4 (-0.4%) | 3.2 |
| LMDZ-INCA | 3092.0 | -4.7 (-0.2 %) | 1.3 | 1.8 | 328.5 | -0.3 (-0.1 %) | 2.5 |
| MOZART-3 | 2926.5 | -14.1 (-0.5 %) | 0.2 | 3.2 | 331.2 | -1.4 (-0.4 %) | 20.3 |
| Model-mean | | -10.25 (-0.4 %) | 2.0 | 2.1 | 330.4 | -0.9 (-0.3 %) | 6.8 |
| **Low cruise (S3)** | | | | | | | |
| EMAC | 3137.3 | +0.1 (+0.0 %) | 3.5 | 1.2 | 340.0 | +0.1 (+0.0 %) | 1.6 |
| GEOS-Chem | 3002.9 | -0.6 (-0.0 %) | 2.2 | 1.5 | 321.9 | -0.0 (-0.0 %) | 2.6 |
| LMDZ-INCA | 3092.0 | +0.8 | 1.6 | 1.4 | 328.5 | +0.1 | 7.4 |

|  | (+0.0 %) |  |  |  | (+0.0 %) |  |
| Model-mean | +0.1<br>(+0.0 %) | 2.4 | 0.9 | 330.1 | +0.1<br>(+0.0 %) | 3.9 |

**Table A4: Summary of mean background halogen mixing ratios in the last 3 years of the baseline (S0) scenario. Units in pptv. Values denoted by – indicate species which are not present in that model.**

| | Surface | | | | 200 to 10 hPa | | | |
|---|---|---|---|---|---|---|---|---|
| | EMAC | GEOS-Chem | LMDZ-INCA | MOZART-3 | EMAC | GEOS-Chem | LMDZ-INCA | MOZART3 |
| Br | 0.0002 | 0.0002 | 0.0001 | 3.0153e-5 | 0.1730 | 0.1252 | 0.1026 | 0.1709 |
| BrCl | 1.9678e-6 | 0.0015 | 5.652e-7 | 4.5890e-7 | 0.5050 | 1.3526 | 0.3435 | 0.4528 |
| BrO | 0.0060 | 0.0028 | 0.0020 | 0.0006 | 2.7007 | 1.8154 | 1.3145 | 2.6226 |
| CFC11 | - | 138.19 | 138.1999 | 248.0824 | - | 59.7229 | 70.9768 | 120.1767 |
| Cl | 1.8753e-6 | 2.0481 | 2.5670 | 3.3410 | 0.0189 | 0.02414 | 0.0213 | 0.0262 |
| $Cl_2$ | 4.9708e-6 | 0.0028 | 1.9384 | 2.6432 | 7.5453 | 5.6160 | 4.3763 | 2.0720 |
| $Cl_2O_2$ | 1.7003e-11 | 1.243e-9 | 7.8370 | 2.1778 | 4.4737 | 18.0780 | 8.6143 | 13.3571 |
| $ClNO_2$ | 3.2010e-8 | 0.0706 | 0.0005 | - | 0.0303 | 0.0050 | 0.1487 | - |
| $ClONO_2$ | 0.02210 | 0.0106 | 0.2481 | 0.2723 | 201.1863 | 265.0219 | 197.7726 | 309.8003 |
| HBr | 0.1244 | 0.01407 | 0.1085 | 0.0131 | 0.4284 | 0.1884 | 0.1691 | 0.2867 |
| HCl | 1.4150 | 3.2963 | 2.5572 | 1.6588 | 669.4262 | 680.3545 | 561.8869 | 936.3695 |
| HOBr | 0.0806 | 0.0116 | 0.0113 | 0.0032 | 1.0023 | 0.8293 | 1.1495 | 1.1218 |
| HOCl | 0.1142 | 0.0973 | 0.0587 | 0.0626 | 9.3144 | 20.2350 | 10.6867 | 8.4008 |
| OClO | 4.5830e-5 | 7.906e-5 | 0.2534 | 4.2968e-7 | 1.7333 | 1.9323 | 1.8834 | 2.1081 |

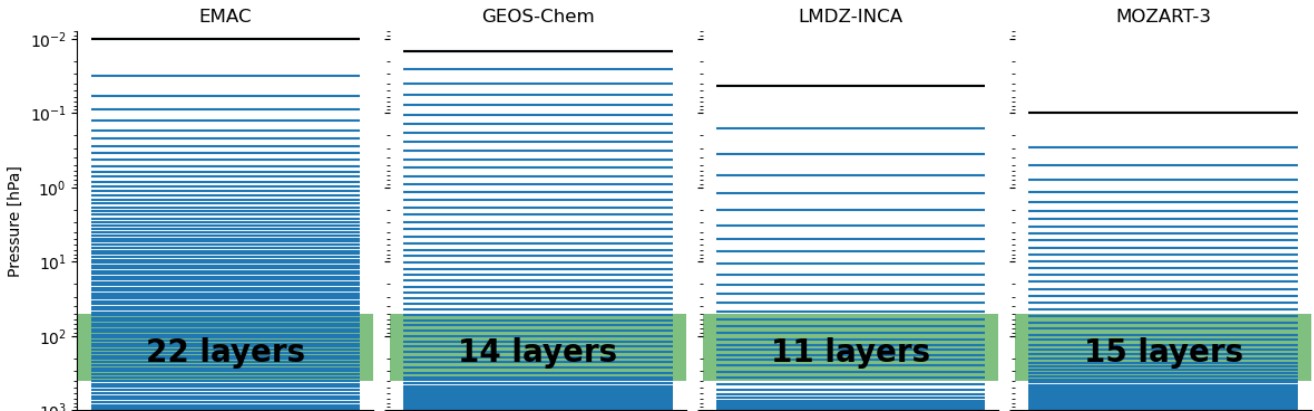

**Figure A1: Comparison of the vertical grid of the EMAC, GEOS-Chem, LMDZ-INCA, and MOZART-3 models. The green region denotes the region between 400 and 50 hPa, which is important to the stratospheric-tropospheric exchange. The count of model layers within this region is shown in the figure.**

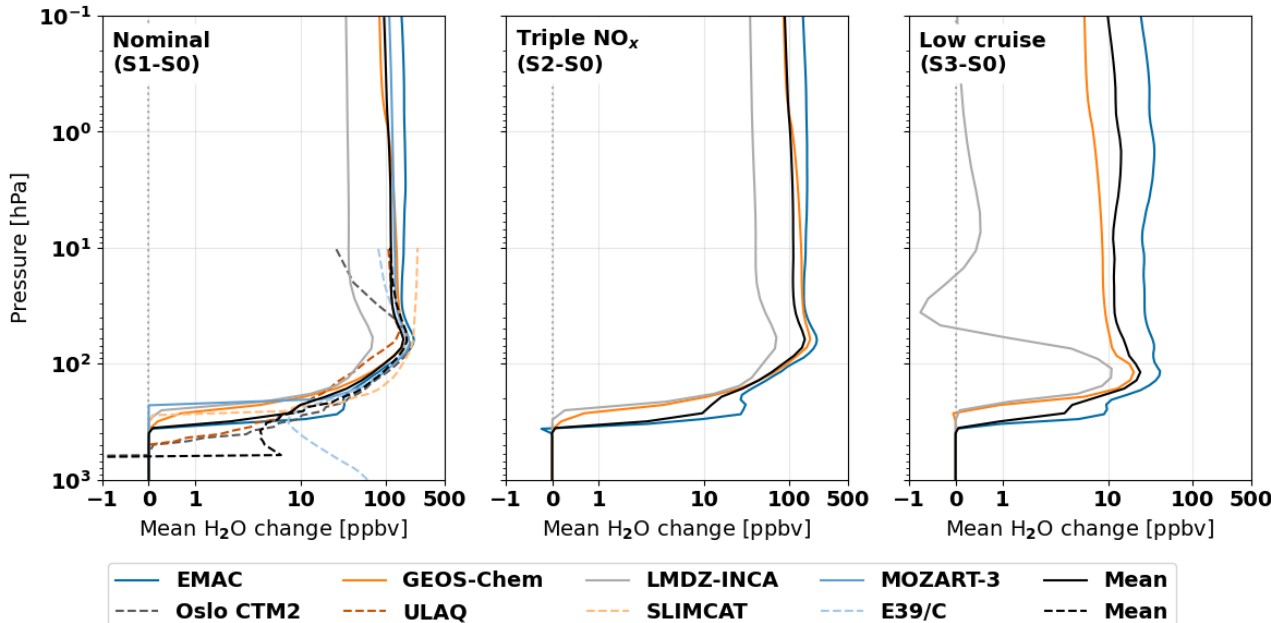

**Figure A2: Mean changes in water vapour mixing ratio over altitude for the nominal supersonic (S1, left), triple NOₓ (S2. middle) and low cruise (S3, right) emission scenarios. Entries with dashed lines are from data from Grewe et al. (2007).**

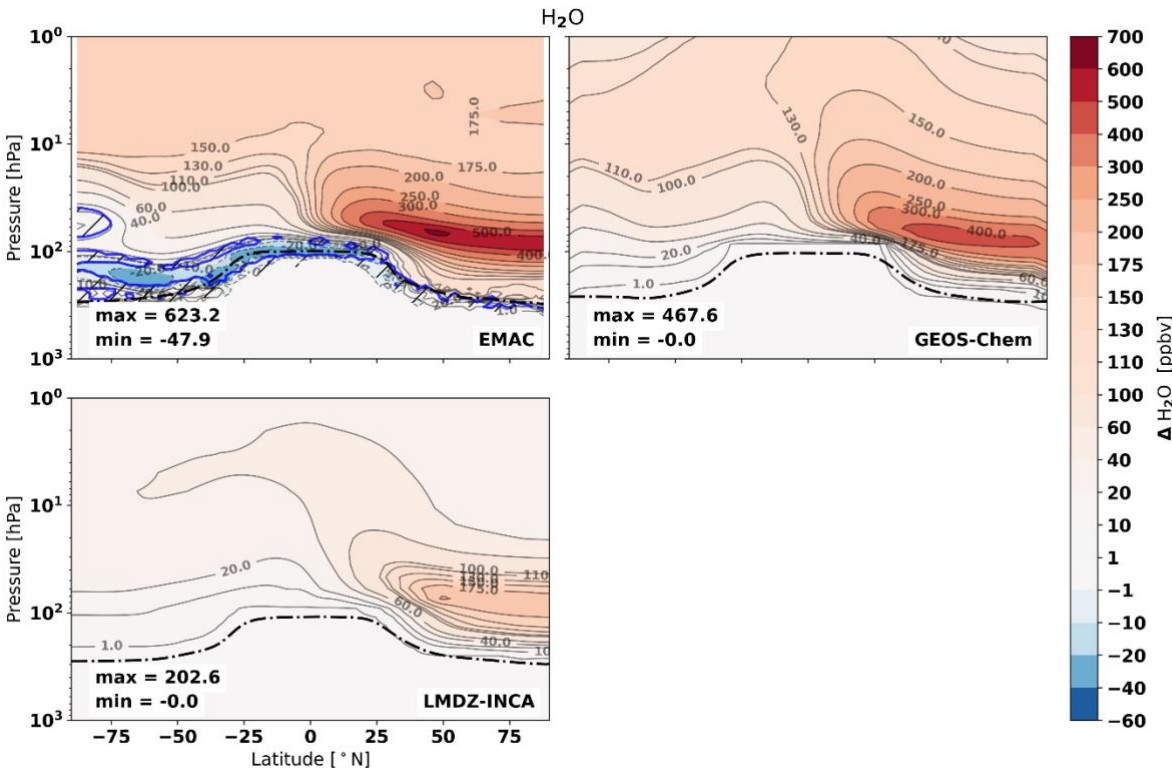

**Figure A3: Changes in H2O volume mixing ratios for the triple NO$_x$ (S2) emission scenario. Hatched areas enclosed by blue lines indicate regions which are not statistically significant for the EMAC results. Dash-dotted lines show the mean tropopause pressure, calculated per model.**

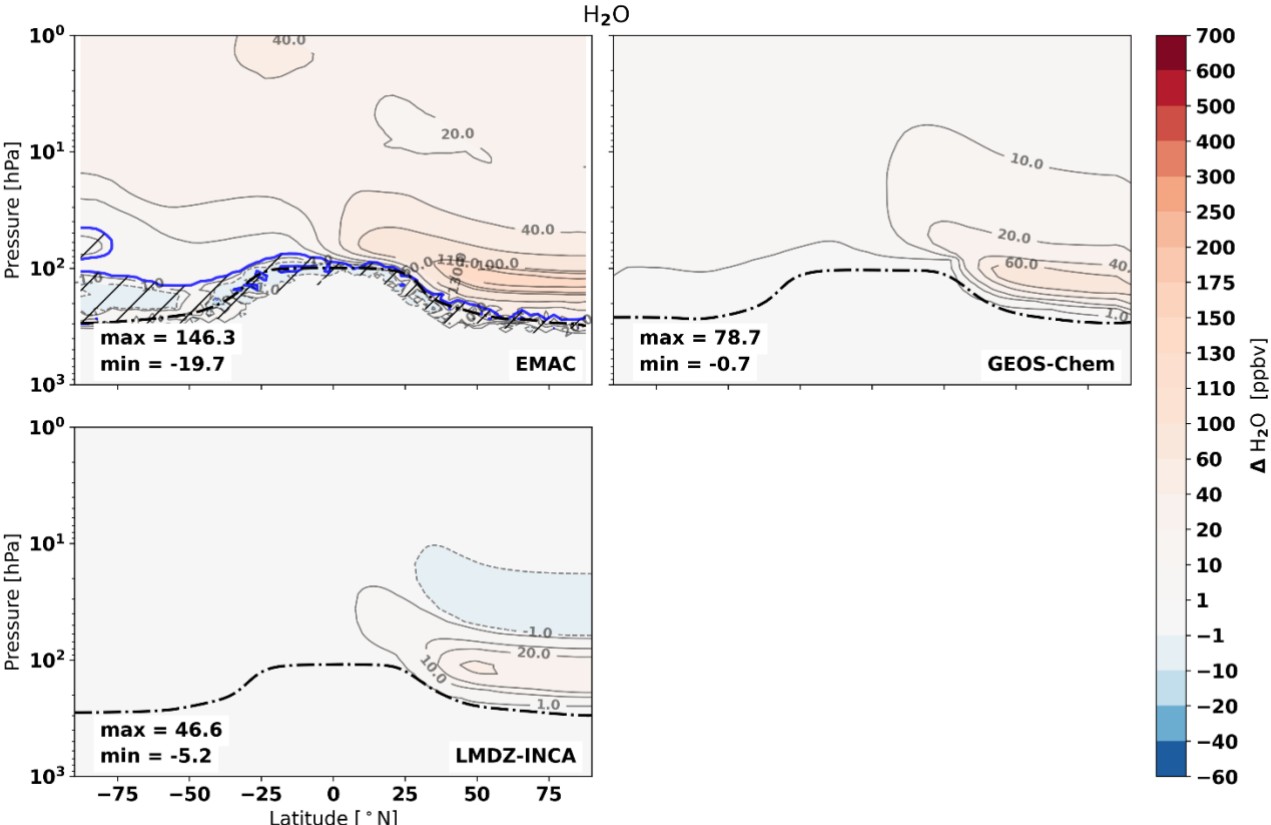

**Figure A4: Changes in H2O volume mixing ratios for the low cruise (S3) emission scenario. Hatched areas enclosed by blue lines indicate regions which are not statistically significant for the EMAC results. Dash-dotted lines show the mean tropopause pressure, calculated per model.**

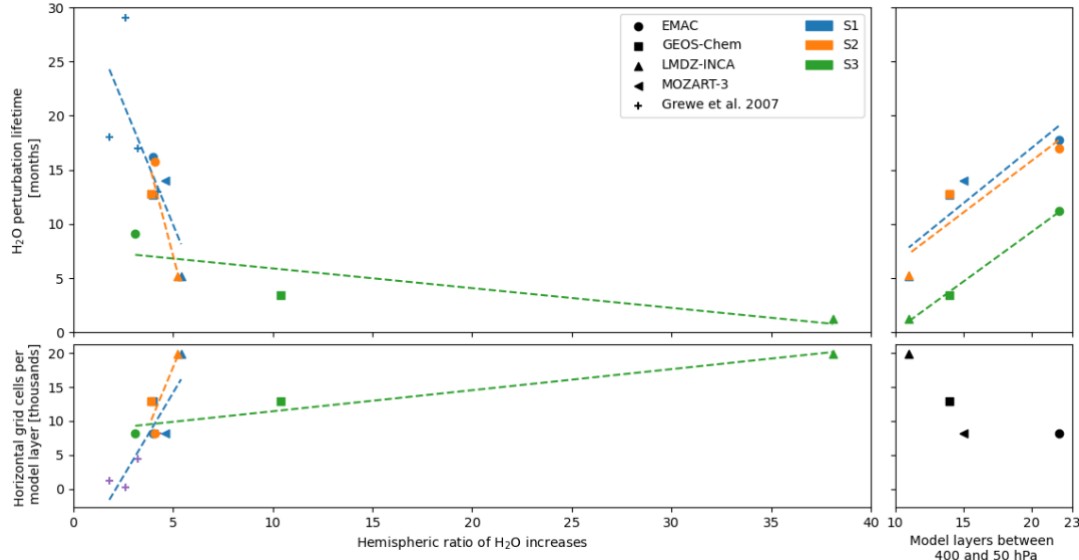

**Figure A5: Comparison of the H₂O perturbation lifetime in months and the hemispheric ratio of the H₂O increases of the nominal supersonic scenario (S1) with vertical and horizontal model grid characteristics. The top left figure shows the relationship between the perturbation lifetime and the hemispheric ratio, the top right the perturbation lifetime and the number of grid layers between 400 and 50 hPa, and the bottom left shows the hemispheric ratio and the horizontal grid fidelity. Markers denote the different models and colour different scenarios. Results from Grewe et al. (2007) are included for their S1-equivalent SST scenario (S5 in Grewe et al. (2007)).**

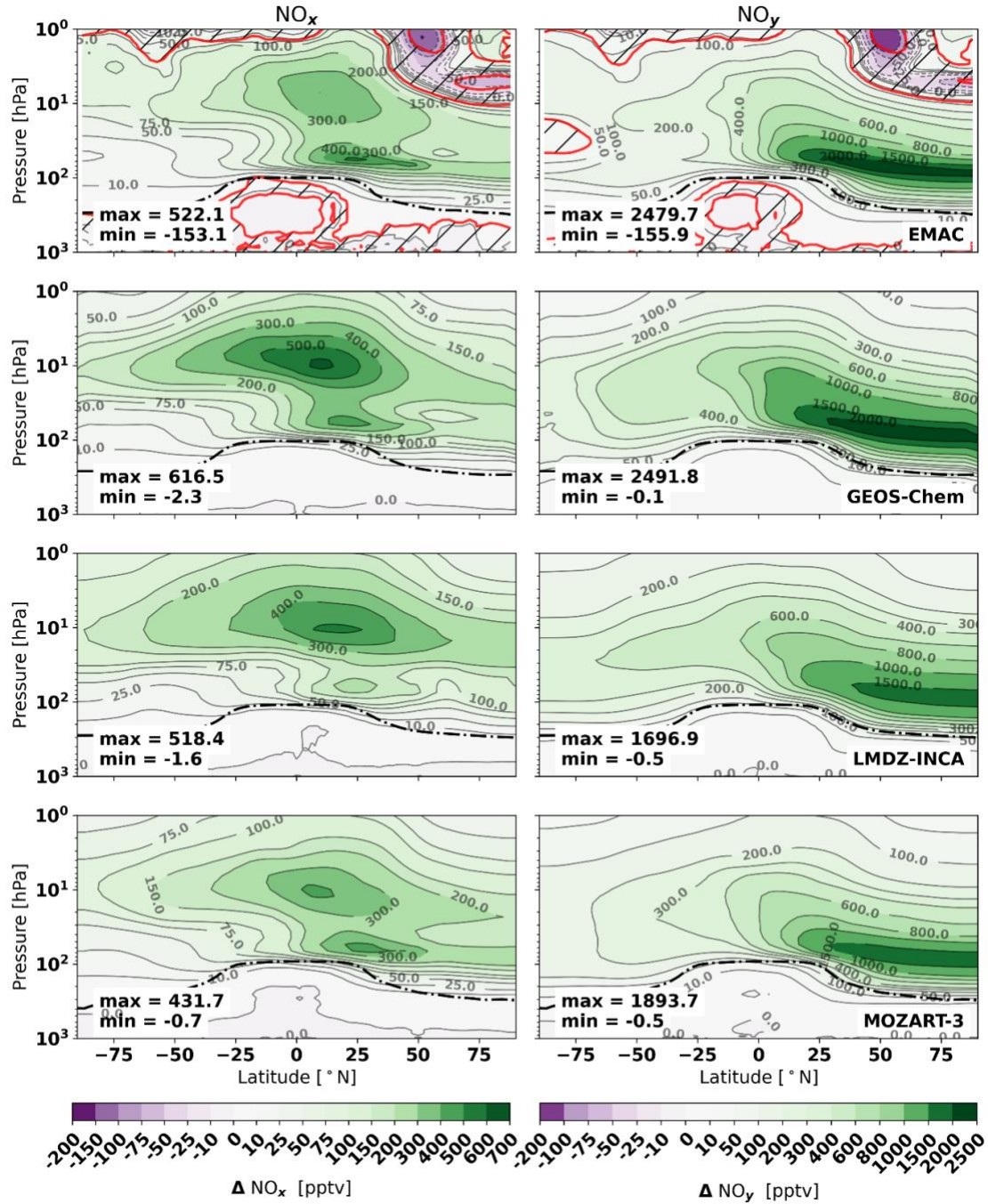

**Figure A6: Mean changes in NO$_x$ (left) and NO$_y$ (right) concentrations in the triple NO$_x$ (S2) emissions scenario across the models. Hatched areas enclosed by red lines indicate regions which are not statistically significant for the EMAC results. Dash-dotted lines show the mean tropopause pressure, calculated per model.**

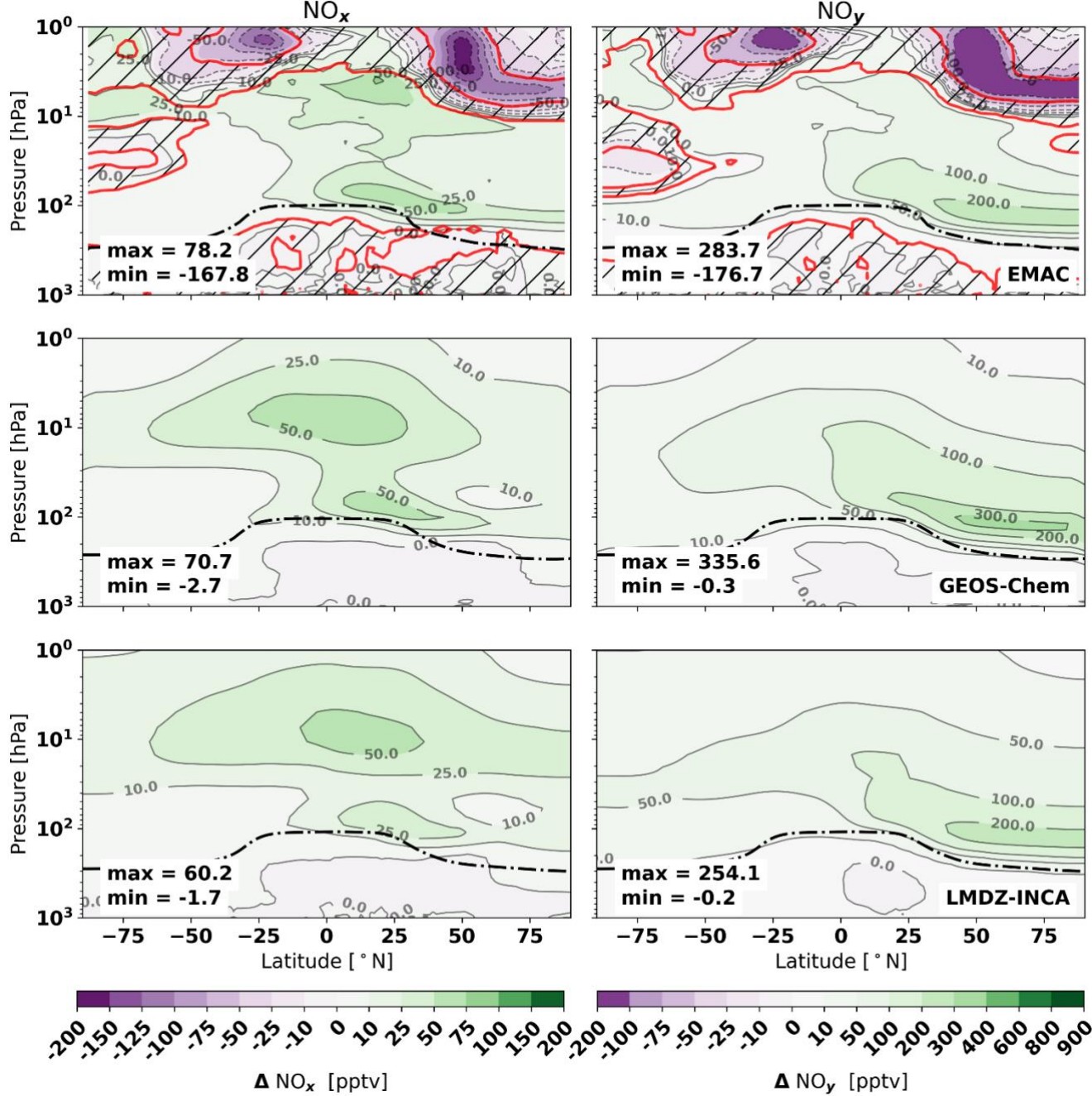

**Figure A7: Same as Figure A6, but for the low cruise (S3) scenario. Hatched areas enclosed by red lines indicate regions which are not statistically significant for the EMAC results. Dash-dotted lines show the mean tropopause pressure, calculated per model.**

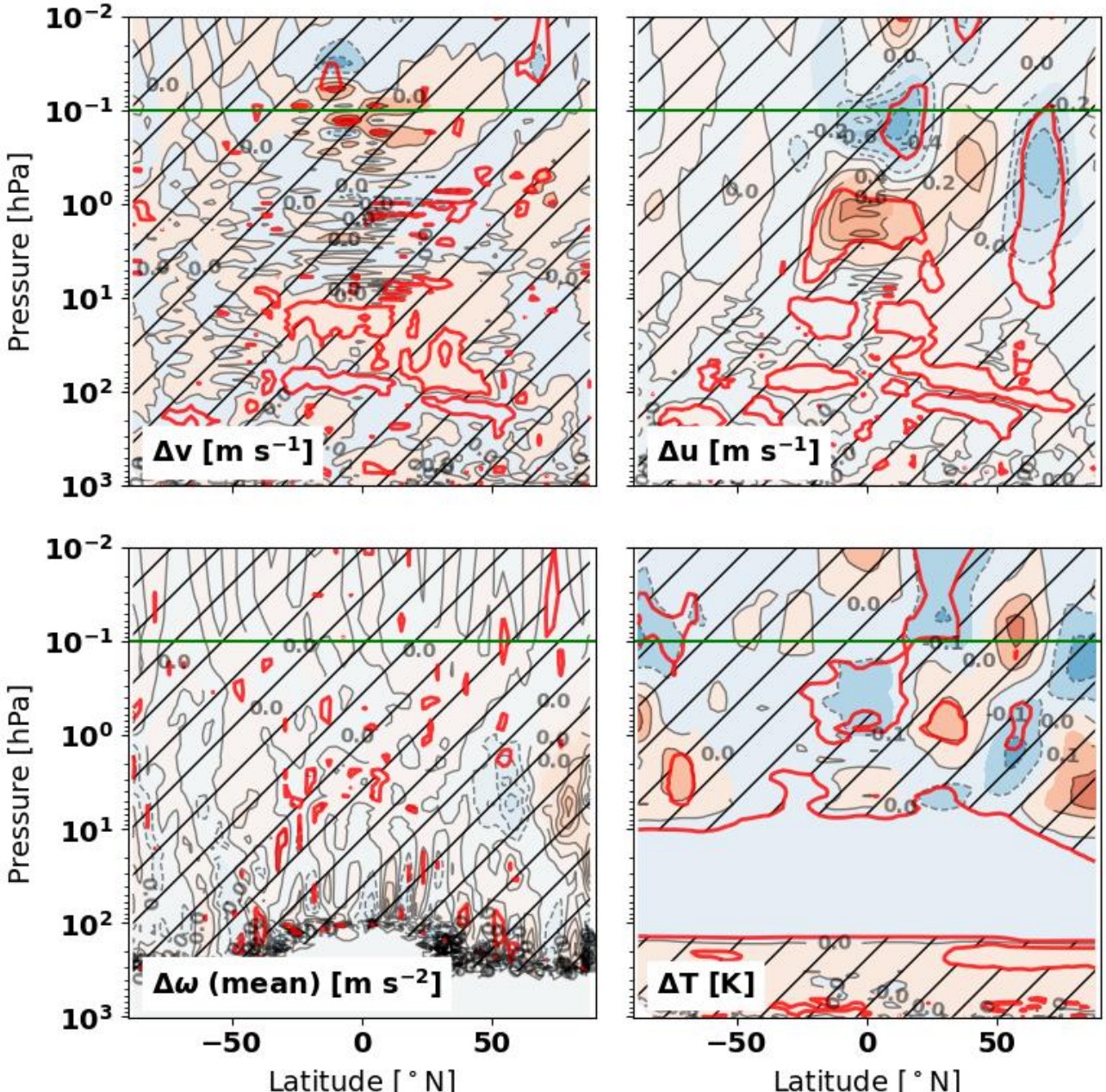

**Figure A8: Changes in mean wind speeds (v,u,ω) and temperature (T) in EMAC in response to the nominal supersonic emissions scenario (S1). Hatched areas enclosed by red lines are not statistically significant over the 6 year evaluation period. Positive changes in u indicate increased eastwards velocities, the positive direction in v is northward, whereas Δω shows changes in vertical acceleration.**

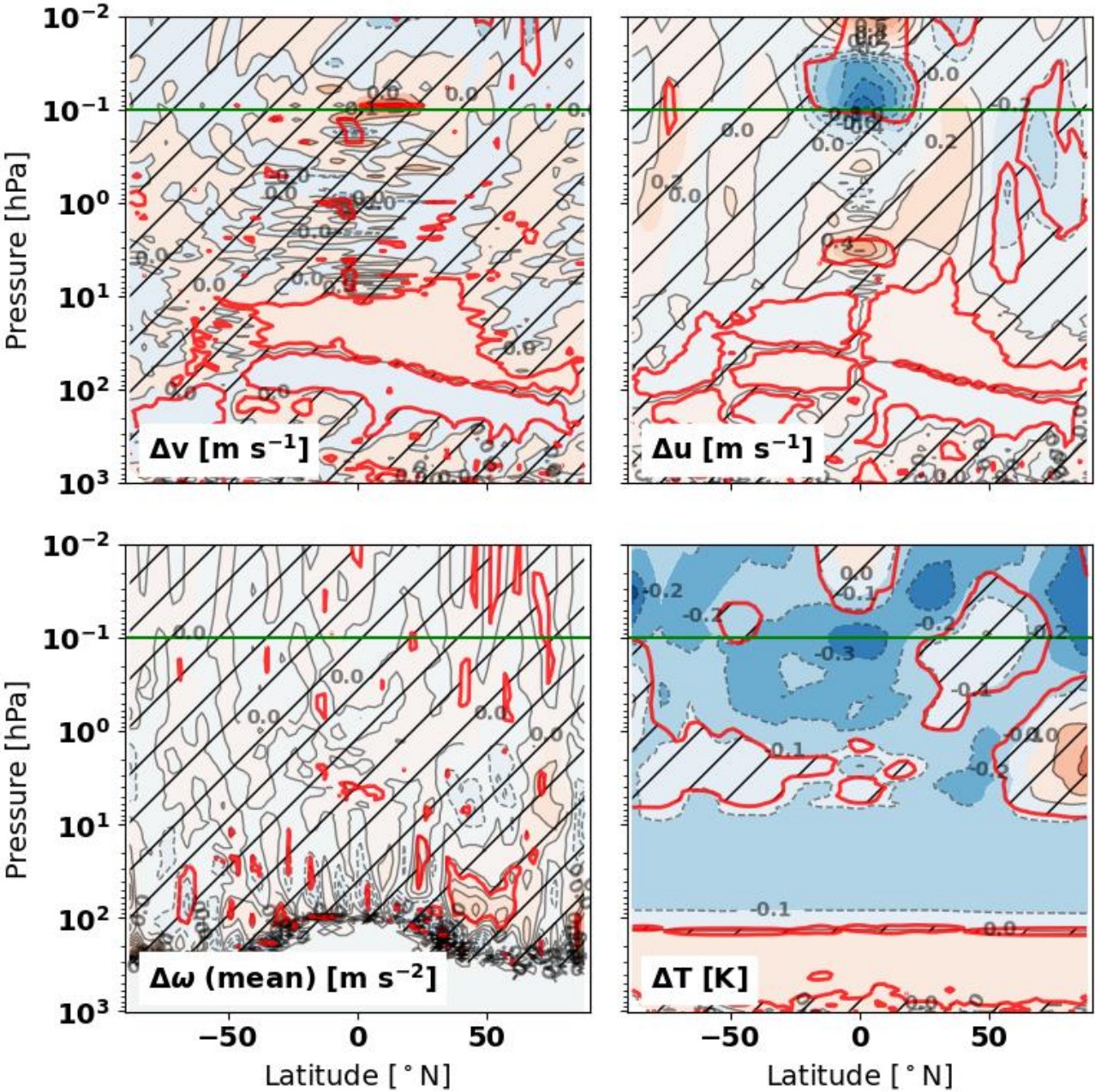

**Figure A9: Same as Figure A8, but for the triple NOₓ emissions scenario (S2).**

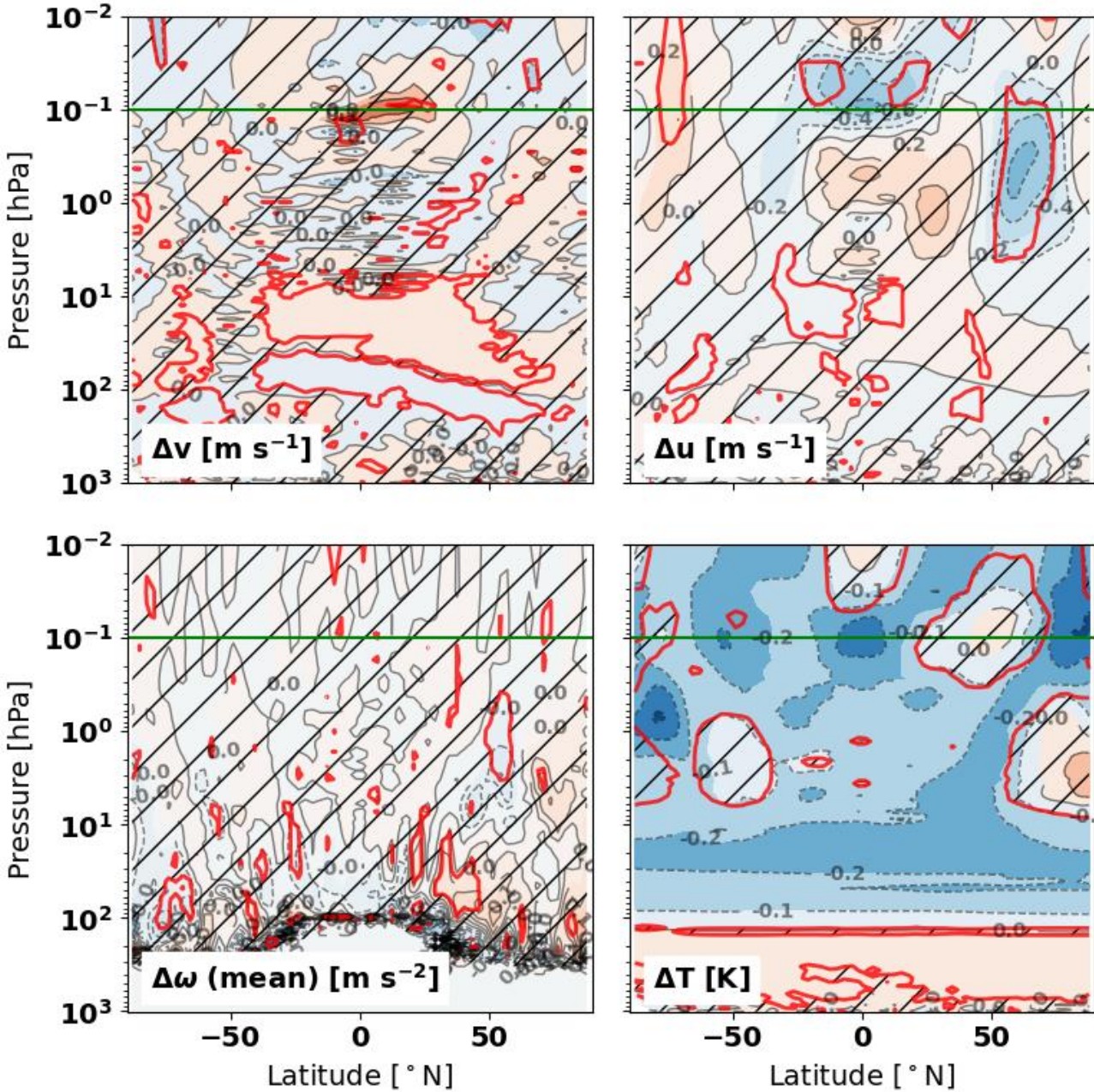

**Figure A10: Same as Figure A9, but for the low cruise scenario (S3).**

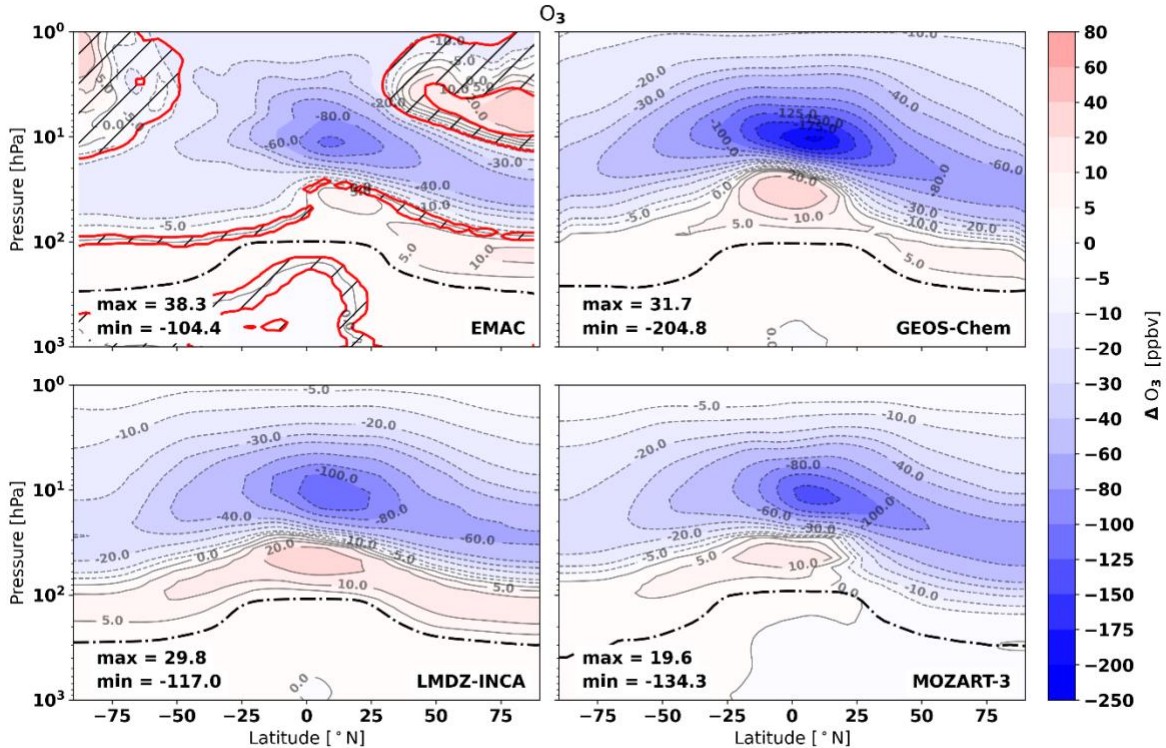

**Figure A11: Changes in ozone VMR for the triple NOₓ (S2) emissions scenario. Hatched areas enclosed by red lines indicate regions which are not statistically significant for the EMAC results. Dash-dotted lines show the mean tropopause pressure, calculated per model.**

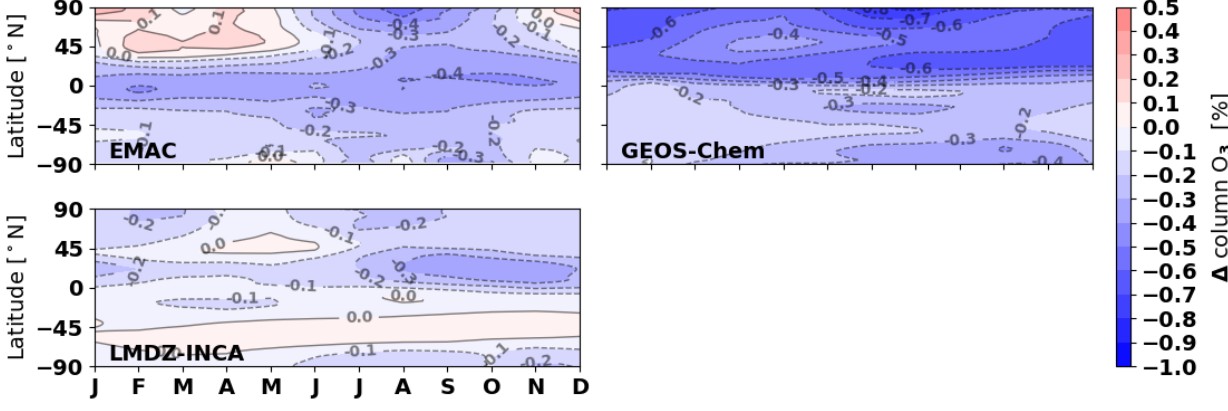

**Figure A12: Similar to Figure A11, but for the low cruise (S3) scenario. Hatched areas enclosed by red lines indicate regions which are not statistically significant for the EMAC results. Dash-dotted lines show the mean tropopause pressure, calculated per model.**

**Figure A13: Mean monthly changes in ozone columns (in percentage) in response to the triple $NO_x$ emissions (S2 – S0).**

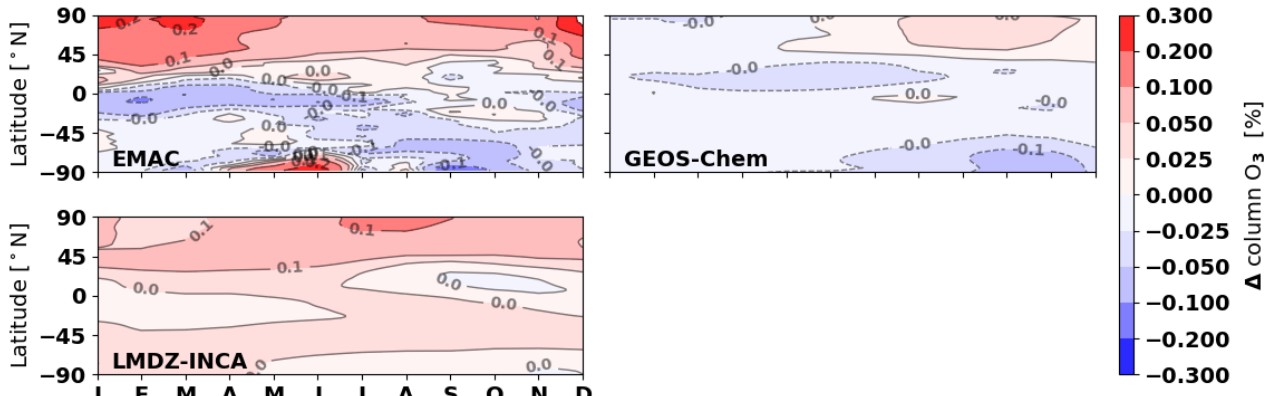

**Figure A14: Similar to Figure A13, but for the low cruise (S3) scenario.**

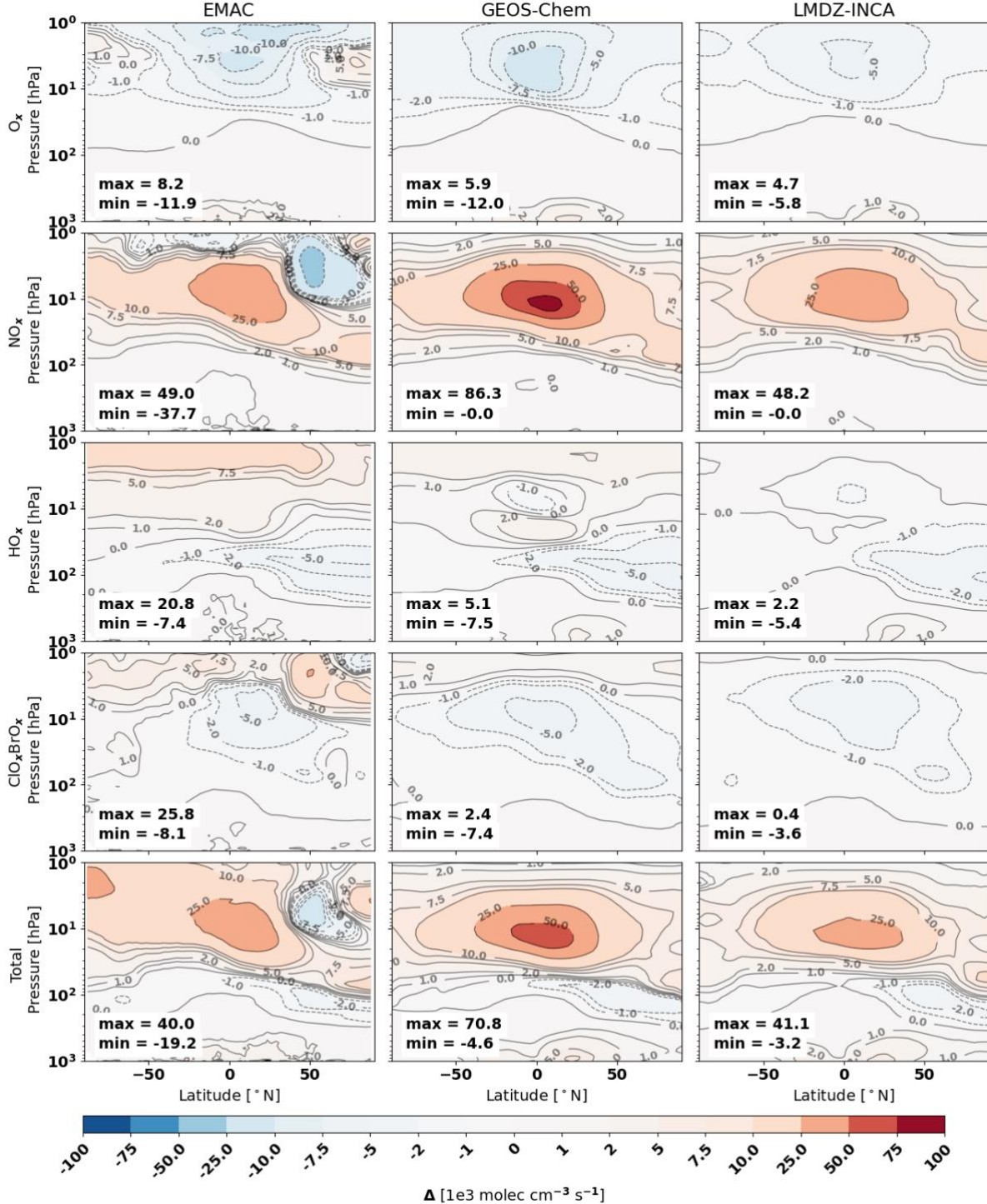

**Figure A15: Same as Figure 8 but for the triple NO$_x$ (S2) emissions scenario**

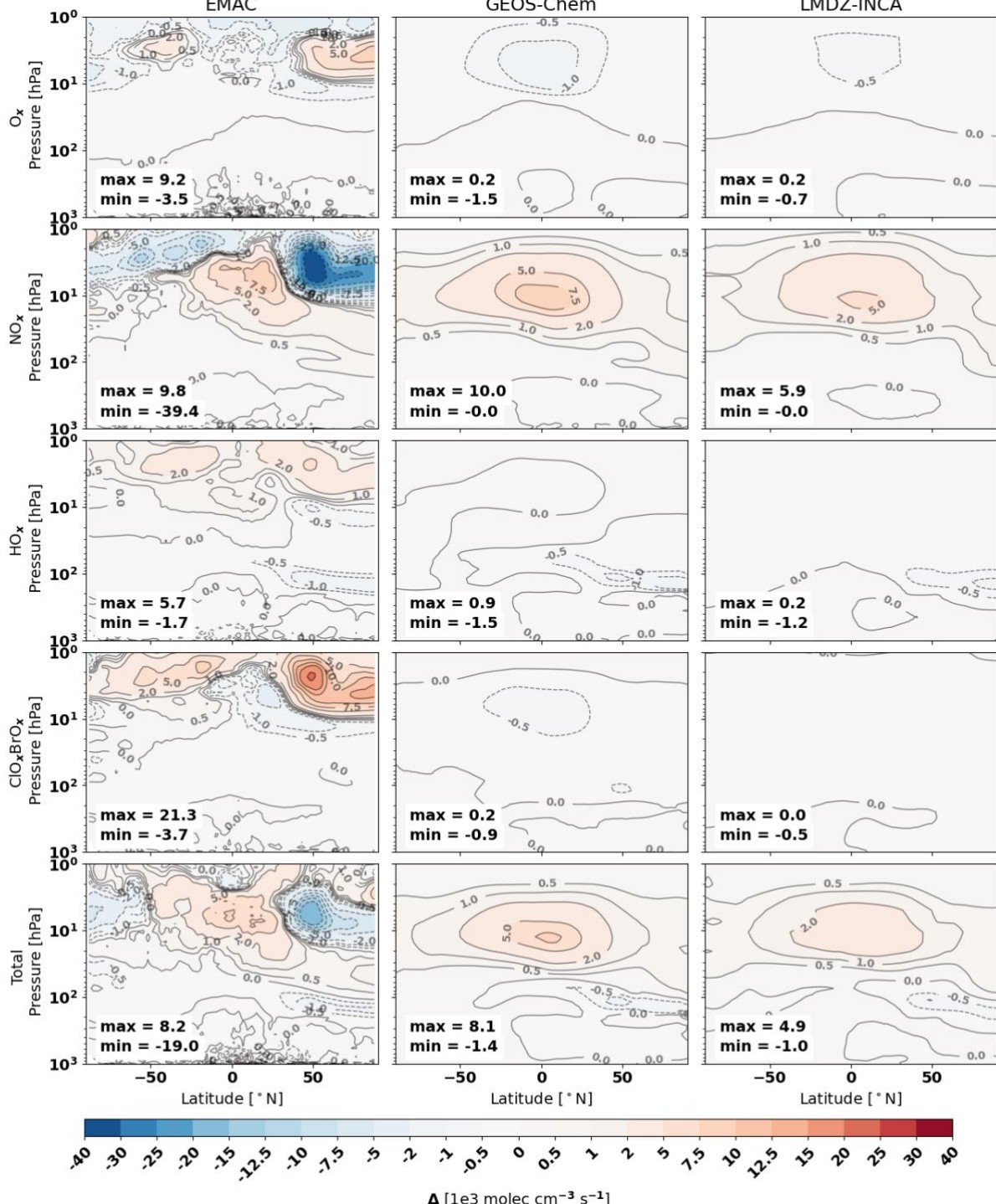

**Figure A16: Same as Figure 8 but for the low cruise (S3) emissions scenario**

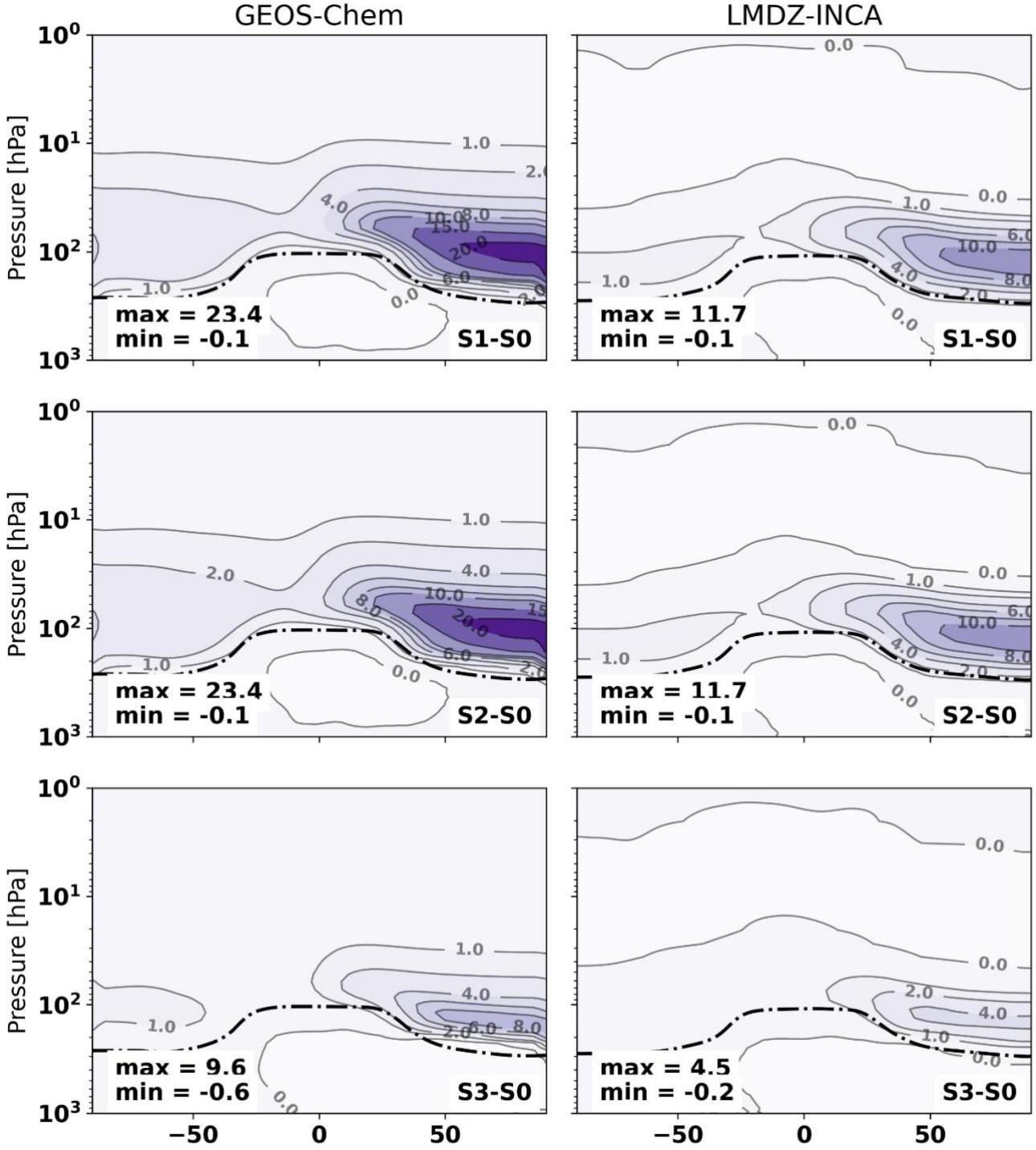

**Figure A17: Comparison of the black carbon aerosol perturbations in $10^{-2}$ ng / m³ for GEOS-Chem (left) and LMDZ-INCA (right).**

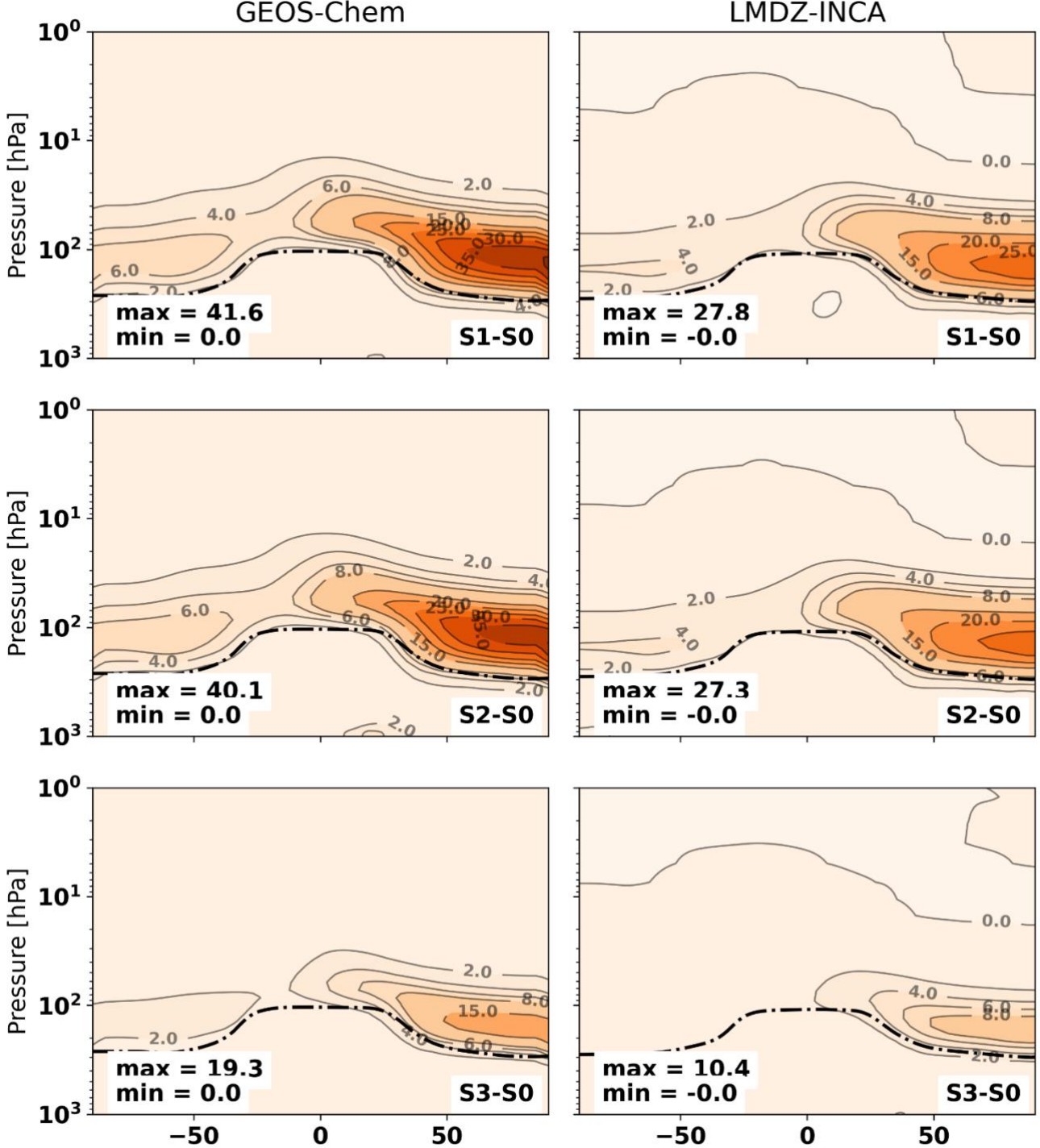

**Figure A18: Comparison of the SO₄ aerosol perturbations in ng / m³ for GEOS-Chem (left) and LMDZ-INCA (right).**

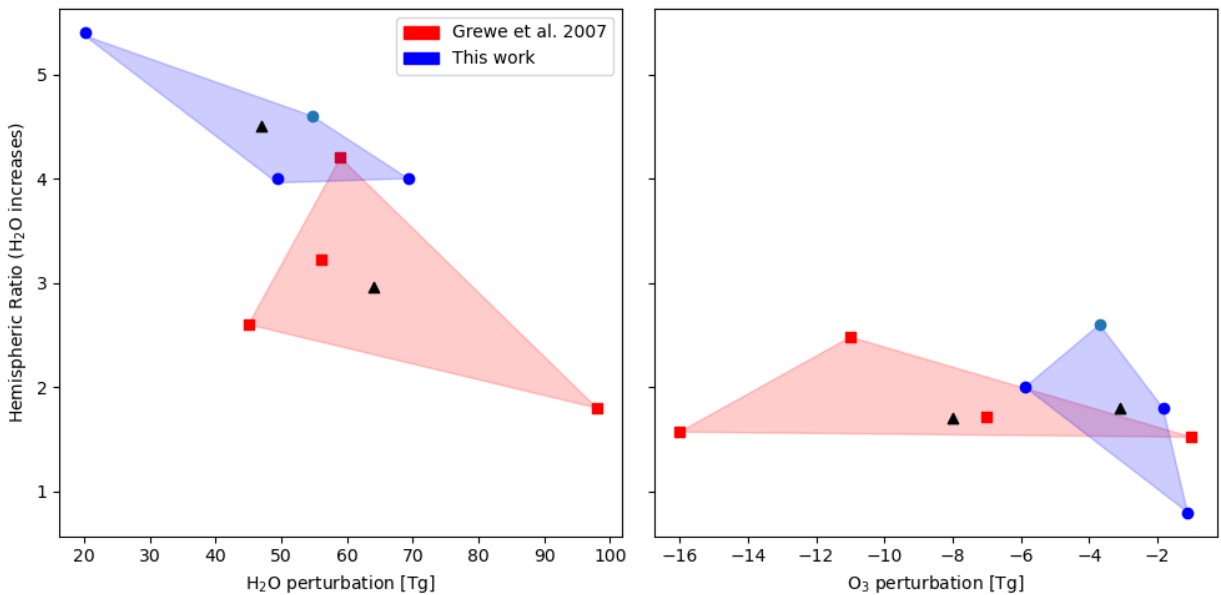

**Figure A19: Comparison of the H2O (left) and O3 (right) perturbations and hemispheric ratios for the models used in this work and Grewe et al. (2007). Black triangles represent the multi-model means.**