# Peer review of "Multi-model assessment of the atmospheric and radiative effects of supersonic transport aircraft"

_EGUsphere, 2024_

## Author Comment (AC1)

Dear Editor and referees,

Thank you for arranging review of our work. We have addressed the reviewer feedback, and we believe this has considerably improved our work and its presentation. Below, you will find our response to the reviewers, answering each comment and detailing how their suggestions and feedback are incorporated in the revision (reviewer comments in *italics*, our response in **bold**, with page/line numbers referring to the updated manuscript).

Overall, we have performed additional simulations to assess the effect of supersonic aircraft emissions on radiative forcing for all models, in addition to the effect on atmospheric composition previously presented. The title of the manuscript has been revised to reflect this wider scope of the work. We have also expanded on the implications of our results for supersonic aviation sustainability decision-making and added a further comparison to conventional subsonic aviation. Additionally, we have restructured the paper to concentrate the discussion elements in a separate Discussion section in order to improve the readability of our work and its presentation.

Thank you again for considering our submission to *Atmospheric Chemistry & Physics*, and we look forward to hearing from you in due course.

Kind regards,

Jurriaan van 't Hoff, Didier Hauglustaine, Johannes Pletzer, Agnieszka Skowron, Volker Grewe, Sigrun Matthes, Maximilian Meuser, Robin Thor, and Irene Dedoussi

**Referee RC1:**

*Van 't Hoff et al. presented a multi-model analysis of the impact of a future supersonic aircraft fleet on upper troposphere and stratospheric composition. Overall, it is a good paper and the results provide valuable information for future consideration of deployment of commercial supersonic aircraft. However, I have some major comments, as well as some minor comments, that need to be addressed before the paper is accepted for publication. Here are the specifics.*

**Thank you for your positive feedback.**

Major comments:

*Section 2.2. The vertical resolution of these models in the upper troposphere and lower stratosphere, could be a critical configuration setup in understanding the aviation impact differences, e.g. Table 3 and Figure 2. Could you elaborate the vertical resolution of each model? For example, how many layers in the critical region (200 hPa – 50 hPa), average layer thickness?*

**Indeed, the vertical resolution is a critical part of the model configurations, which was not fully explored in the initial manuscript. We have made several revisions to improve upon this: Table 2 has been amended to explicitly compare the number of vertical model layers between 400 and 50 hPa (we extended the critical range to 400 to consider the tropopause levels at their lowest as well) , and the average layer thickness in this range is compared in text (Lines 160-161, 188, 216, and 238). In the appendix we have added Figure A1, which directly compares the vertical model grids, and the vertical layer count in the critical region is now integrated further into relevant discussions of perturbations (e.g. lines 327-350).**

*One of the most interesting findings from this work is the distinctive responses in NOx, O3, and H2O in EMAC which is an online vs the other three models with nudged meteorology. As the authors mentioned in the text, this is likely due to the meteorological/dynamical feedback induced by the composition changes in H2O and O3. One major weakness of this work is that while the authors have presented many figures showing the chemical responses both in the main text and in supplement, I couldn't find any showing the dynamical changes, e.g. temperature, U/V/omega, other transport terms, in EMAC due to supersonic aircraft emissions w.r.t. the control run. It would be useful to include these results as they will help to elucidate the changes in H2O, NOx, NOy, and ozone in different parts of the atmosphere.*

**Thank you for this feedback. We have moved the figure showing the changes in temperature fields to the main text of the manuscript (Figure 5), and added a paragraph for explicit discussion of the changes in dynamics (lines 371-385). Furthermore, we have added 3 new figures to the appendix (Figures A8-A10) to show the changes in U,V, and omega alongside the temperature fields in response to each of the emission scenarios. With regards to the wind fields, we have found that the**

**majority of changes are not statistically significant due to the model's nudging of these terms. This is why we have chosen to add these to the appendix over the main text, but the main text is updated to explain this (lines 386-387).**

*Section 3.6. This is a good discussion section, but I would suggest, instead of "modeling consideration", make it a more comprehensive discussion to include the following aspects: i) what are the key findings from this paper? Ii) how these findings agree/differ from previous work (Grewe et al., 2007; Zhang et al., 2023; Eastham et al 2022; Matters et al., 2022), qualitatively and quantitatively? iii) what are the implications in differences in offline CTMs vs. online meteorology (as in EMAC)? For example, does the results in this work suggest that models with offline meteorology are possibly not the optimal way to assess supersonic aircraft emissions impact as the offline setup is inadequate in capturing full responses. Or alternatively, you can state what are the key impacts that can be captured by offline CTM and what are the key impacts that occur in atmosphere but missing in offline CTM representation.*

**Based on this suggestion, and a similar comment of RC2, we have revised the manuscript and updated the "Modelling considerations" subsection to a comprehensive Discussion section instead (Section 5). We have also concentrated all comparisons with literature in this new section and strengthened the comparison of the offline and online models in this discussion. This section has been expanded to also discuss the sustainability implications of our results for supersonic civil aviation.**

*Other than the technical documentation of composition changes, what is the bigger implication of this study? Can the authors add a few sentences in the conclusion section about the implication? As presented throughout the paper, the adoption of A1 scenario supersonic aircraft fleet leads to very small changes in H2O (1-2%), NOx (1-3%), O3 (a few tenths percent, <1DU). The A3 lower cruise scenario impact is even smaller. Does this suggest it is reasonable to consider supersonic aircraft fleet in the future?*

**Thank you for this feedback. In response to this, and similar feedback from RC2, we have made several revisions to the manuscript. We have added assessments of changes in radiative forcing from all four models to the manuscript during the revision (Section 4.6). With these radiative assessments we also facilitate a new comparison with subsonic aviation (lines 603-623). The new results highlight that both the $CO_2$ and non-$CO_2$ driven radiative impacts increase considerably per passenger kilometre when they are flown by one of our supersonic aircraft concepts rather than subsonic aviation, even in the lower-cruise scenario. Furthermore, we have added additional discussion reflecting on the difficulty of resolving the impact disparity between supersonic and subsonic aircraft, and further implications on human health (lines 625-636). We believe that these new additions paint a much better picture of the big-picture of the adoption of supersonic aircraft, compared to what was previously only communicated through changes in composition.**

Minor comments:

*Title and elsewhere in the text. It is more common to use "multimodel intercomparison" instead of "intermodal comparison". Consider revise.*

**Thank you for the suggestion, considering the addition of radiative assessments we have updated the title of the work to "Multi-model assessment of the atmospheric and radiative effects of supersonic transport aircraft".**

*L17 & L96. Other than EMAC, I would not agree to use the word "state-of-the-art" to describe the remaining three models, especially in the context of this paper here. The CTM capability does not allow to examine the meteorological feedback of the composition change induced by the aircraft emissions. You can use a softened word such as "comprehensive atmospheric models".*

**The wording has been updated following this suggestion (lines 16, 102).**

*L19-20. Here and in most of the places through the text, you are presenting the ozone changes in % unit. It would be much more helpful to me if you express these changes in dobson unit (DU), or at least include the DU changes in parenthesis, which is the common unit used in literature.*

**We adopted % as a standard for the comparison to account for the differences in the models' baseline $O_3$ columns, however, we agree that DU changes may be more valuable for many readers.**

**We have revised the manuscript to add changes in DU alongside the changes reported in percentages (Changes throughout section 4, the abstract, and conclusions).**

*L28. While it isn't wrong to use "intercontinental transport", to not be confused with intercontinental transport of air mass or pollutants (which is another major topic in the composition community), I would suggest you use "intercontinental transportation" instead.*

**Wording has been adjusted to use "transportation" (Line 27).**

*L141. Should be Burkholder et al.*

**Typo has been corrected, thank you (Line 149).**

*L196-199. Please clarify what is the physical water vapour tracer vs. full water vapour tracer, how are they treated differently in the model.*

**The "physical" water vapour tracer refers to the water vapour tracer used by the LMDZ general circulation model, which is not subject to chemistry but is affected by physical processes such as transport, phase changes, and cloud formation amongst**

others. The "full" water vapour tracer is the H$_2$O implementation in the INCA model, which is also subject to (photo)chemistry as well as chemical sources and sinks. Below the model tropopause the INCA tracer is prescribed by the LMDZ tracer, whereas in the stratosphere the INCA tracer evolves freely subject to transport and chemistry.

**We have revised the paragraph in question to provide a simpler explanation by drawing parallels to the other models (lines 207-209).**

*L208. Missing parenthesis after 2011.*

**Thank you, corrected (line 227).**

*L215. What do you mean by "fixing it the troposphere"? Relaxed to climatology, or nudge with ERA-Interim data?*

**With "fixing the troposphere" we referred to models that prescribe tropospheric mixing ratios of H$_2$O.**

**The line in question has been reworded (lines 233-234)**

*L237. Change "Firstly" to First. Also if you are listing First, shouldn't you also use Second and Third, etc. ? I don't see any in the following sentences.*

**The wording has been adjusted, "Firstly" has been removed entirely (line 261).**

*L238-239. This sentence seemed to be out of place. Probably better suited somewhere in Section 2.2.*

**Thank you for the suggestion, the content has been moved to a separate section ("3.5 Evaluation method")**

*L245 and thereafter – the use of term "hemispheric fraction". As you defined in the text, this is the ratio of the perturbation between the two hemispheres, not the fraction. By mathematical definition, a fraction is a numerical quantity that represent the portion/part of the whole thing and varies between 0.0 and 1.0. Please use a more accurate term to describe this.*

**The term "hemispheric fraction" has been replaced by "hemispheric ratio" throughout the manuscript.**

*L247. Can you actually learn about interhemispheric transport, or is it just the NH --> SH transport since aviation emissions are pre-dominantly emitted in the NH, particularly in the NH mid-latitudes?*

Thank you for this question, around 90% of the supersonic aviation emissions are indeed in the northern hemisphere, so it is correct that we may evaluate northern to southern hemispheric transport more than the hemispheric mixing.

The manuscript is revised to more accurately reflect this. It no longer suggests that hemispheric mixing is assessed directly, instead we focus on comparing the ratio of the stabilized perturbation versus the hemispheric ratio of the emissions (added in lines 272-273). Per the comment about line 257, the link to perturbation lifetimes is also integrated (Lines 345-348).

*Section 3.1. I find this section is a little hard to follow as the authors are frequently hoping back and forth from model mean responses to inter-model differences, and vs. previous results from Grewe et al. (2007). May be consider re-arrange the discussion in the following order: model mean impact, including H2O, NOx, Ozone in A1 scenario, A2 scenario and A3 scenario--> how do the A1 results compare with Grewe et al. (2007). You may want to mention model spread here, but I find it distracting if you discuss the details of why one model is different from the others. You can move the discussions into the later sections.*

In the revised manuscript, all comparison of our results with other literature is concentrated into the new comprehensive discussion section (Section 5). We believe this should resolve the feedback given by the reviewer.

*L 257. I am not convinced that the authors presented conclusive evidence to suggest that a smaller NH/SH ratio is due to reduced hemispheric exchange. Did you check the changes in H2O lifetime? Short H2O lifetime could lead more faster removal of H2O (from aviation emissions) when it is en route from the NH to SH? There are other possible explanations as well.*

Thank you for this feedback. The reviewer is right that there are more factors that could affect these results. As the reviewer suggests, we also find a clear trend between the $H_2O$ lifetime and the hemispheric ratio. We expect that the discrepancy in lifetimes is more influential than any differences from the horizontal grids, and have revised the manuscript accordingly.

Specifically, the text has been revised to better describe our findings, considering the link between the perturbation lifetime and hemispheric ratio. We have also added a new figure to the main body of text to show this link and the relation with model resolutions (Figure 3, the previous Figure 3 has been moved to the appendix as figure A2). Per the previous comment about line 247, explicit mentions of the hemispheric exchange have also been removed. This is now collected in a new discussion supporting figure 3 (lines 327-350).

*L277. What is the reason behind a net-decrease in H2O in the SH? Is it due to temperature perturbation? Is it due to transport responses from the composition changes? You may consider analyze the model temperature fields, vertical and horizontal transport terms to understand the causes.*

Unfortunately, we were unable to pinpoint the cause of this phenomenon with sufficient certainty after further investigation. We originally hypothesized that it is related to the dynamical feedbacks due to the coupling of EMAC's $H_2O$ tracer to relative humidity, however we cannot identify statistically significant changes in dynamics in this regime.

The net-decrease in southern hemispheric $H_2O$ mass was previously calculated from only a partial timespan of the EMAC data (3 years). The results in question have been updated to the extended EMAC dataset (6 years), where we no longer find a net-loss of $H_2O$ in the southern hemisphere. We therefore expect that the net-decrease in southern hemispheric water vapour may have been the result of the model's internal variability, and since this is resolved in the longer average, we omit this line from the revised discussion.

*Figure 2. Are the dash-dotted lines model tropopause from each model? Please clarify in the figure caption.*

They are indeed the mean tropopause pressures for each of the models. The caption of the figure has been adjusted to clarify (Figures 2, 4, 6).

*L312. I am not sure you should refer to the inter-model differences as "discrepancies", large variances may be? By discrepancy, you are implying something is unexpected or some models may be inadequately wrong.*

The wording has been adjusted to use spread between the models instead (Line 327).

*L314. Change "afterwards" to (section xx).*

This line in question has been removed entirely in the revision.

*L315-327. See my comment above on model vertical resolution discussion.*

Discussion of the vertical model resolution has been extended, as described in our response to the relevant comment.

*Figure 3. I would suggest you use the same value range for x-axis on all three panels. This way one can clearly identify the magnitude of differences in H2O changes.*

Thank you for the suggestion, the figure has been revised to incorporate it. Please note that in the revision the Figure 3 of the original manuscript has been replaced, and is now found in the appendix instead (Figure A2). We have also applied this feedback to other figures where possible (Figures A2,7,8).

*L322. Results from model are not Observed! Models are not observations. You may use alternative words such as shown, found, identified, etc.*

We have changed the use of the wording of this line in question (Line 339), and other instances where "observed" was used throughout the manuscript.

*L329. "This is reflected …" What is reflected?  Please be specific.*

This referred to the relationship between the $H_2O$ perturbation lifetime and the (vertical) model grid, during the revision this line was removed.

*L354. Change "fit" to "agree with"*

The line in question has been removed during the revision, as it and the associated paragraph are no longer relevant with the revised results.

*L385. Delete "be".  Also slow down was used twice in this sentence.*

This line has been revised following the suggestions (Lines 384-386).

*L392. As I have stated in previous comments, it would be useful to look at the meteorological changes in EMAC to confirm whether the unique responses in EMAC is due to online meteorology.*

As discussed in the response to the associated major comment, new discussions and figures have been added to the manuscript to evaluate the changes in online meteorology (Figure 5, A8-A10, lines 373-390).

*L400. Change to "These effects combined"*

The line in question was removed in the revision of Section 4.4.

*Figure 8 caption: change "changes in percentage of the ozone columns" --> "changes in ozone columns (in percentage)".*

Suggestion incorporated.

*L475. Delete "also"*

Suggestion incorporated (Line 461).

*L564.  Consider use "internal variability" than "noise".*

Thank you for the suggestion, the suggested wording is incorporated in the revised manuscript (e.g. line 253). The line marked by the reviewer was removed during the revision.

*L547-548.  While I agree it would be nice to have models use similar or save vertical grid/resolution, the adoption of vertical grids is a decision made by each modeling groups in consideration of all factors, including numerical recipes. This*

*recommendation/suggest is just an overkill. I would suggest delete this sentence. As an alternative approach, you may mention that differences in vertical grids can be a major contributing source of stratospheric H2O lifetime.*

**Thank you for your feedback, the statement in question has been altered per the reviewer's suggestion (Lines 580-582).**

**Referee RC2:**

*Van't Hoff et al. explores the effects of an implementation of a supersonic aircraft fleet on atmospheric composition, comparing results from four models and contrasting them to a previous study using different models but a similar emission scenario. The concerns over environmental effects of supersonic aircraft sparked research decades ago and given that these aircraft are now back on the agenda, the study is timely. The study is comprehensive and mostly well-written, but I still have some comments and questions to be addressed before I can recommend publication.*

**Thank you for your feedback and suggestions.**

*I think the reflection on the broader importance of this study can be improved. Many of the changes from implementing a supersonic fleet are provided with two significant digits but are quite small. While I understand quantification of radiative effects may be beyond the scope of the study, but a brief discussion about climate implications and/or e.g. placing the changes from implementation of supersonic aircraft into the context of the current impact of the fleet, would be good. Including the potential role of the reduced Nox emissions at lower altitudes, which I couldn't really see discussed much. Strangely, the authors only mention H2O as a climate driver, not ozone.*

**Thank you for this suggestion, we have adapted the manuscript accordingly, and we believe this has greatly improved the value of our work. In the revised paper we added an assessment of the effects on radiative forcing (Section 4.6), and we utilize this to facilitate a discussion on the climate implications and to perform a comparison with subsonic aviation by calculating the radiative effect of replacing a revenue passenger kilometre flown by subsonic aircraft with our supersonic concept aircraft (lines 603-623). The introduction has also been revised to better introduce the climate implications of NO$_x$ and aerosol emissions (Lines 5-61), and the new results better show the roles of these emissions in the overall climate effect (Sections 4.6, 5).**

*In several places, the authors conclude that there is significant improvement in the model agreement compared to the older study. However, looking at their figures, in particular the vertical profiles, I don't think this is a statement that can be made without further quantification or a definition of what the authors consider improvement. Moreover, given that the comparison of inter-model differences is limited by the fact that there are notable differences in model setup and inconsistent parameterizations, there is even a stated issue with one of the model's treatment of H2O, I'm not convinced this "improved*

*agreement" is something you want to trust. Finally, given that different models are used in the current and the old study, there's a significant limit to how far the comparison of agreement can be pushed.*

**This is a fair assessment; we can understand that the "improvement" discussed in the original manuscript was nebulous without exact definition. Furthermore, it is true that it is hard to assess improvements between sets of entirely different models.**

**In the revised manuscript, we replaced mentions of "significant improvements" entirely with more specific language to describe what is compared. Furthermore, in the revised manuscript we have reduced the direct comparison with the older study, given the difficulty with the substantial model differences, instead we focus more on comparing big-picture implications (Revisions throughout sections 4 and 5). During the revision we also further investigated the MOZART-3 $H_2O$ model, and identified and corrected the underlying issue.**

*The study speaks of substantial improvements in the modeling of ozone chemistry as the reason for the improved agreement, but studies still show significant differences from observations as well as between models in the baseline representation of the atmospheric composition, and there are model differences in the assessment of the current aviation fleet. More complex chemistry does not always equal improved performance. The authors should consider including a (clearer) discussion of the baseline model performance and whether biases play a role in the differences when assessing emission scenarios.*

**Similar to the previous point, we have updated the manuscript to steer away from the previously mentioned "substantial improvements" that cannot be backed up without a more dedicated study of the modelling. In the revised manuscript we have added a comparison of the composition of the baseline atmospheres (Lines 308-315), and any comparisons between ozone (response) modelling have been made more precise and grounded in the results instead. Statements surrounding improvements in ozone modelling have been removed entirely, in favour of clearer direct comparisons or focus on bigger-picture effects.**

*The methods section would benefit from some streamlining and clarifications – see detailed comments below. The results section reports is comprehensive but a bit sloppy at times, and it would help the reader with a bit more precise language, e.g. making sure to say what the increases and decreases are relative to (e.g. "There is little change in terms of the H2O perturbation" – does this refer to change relative to the baseline or change compared to the difference between A1 and baseline), consistent use of +/- or not in front of numbers, etc. I also wonder if the current level of precision of the reported numbers is needed, or even warranted here, given model spread or if it rather distracts from the core message of model differences. Moreover, I think there is some repetition between section 3.6 and preceding sections that could be avoided (by combining the discussion and comparison with other studies into one section), hence improving readability (of a section that is quite long).*

**Thank you for this feedback, we have made several revisions to the manuscript in response to this and the associated specific comments.**

**The methods section has been streamlined and standardized. We have split the methodology section into two focused sections instead (2. Emission scenarios and 3. Atmospheric Modelling), and the model descriptions (Sections 3.1 to 3.4) now follow the same structure and discuss the same components of the model configuration, as the reviewer's specific comments identified differences in these descriptions.**

**Furthermore, we have made revisions throughout the manuscript to use more precise language and to improve consistency in the presentation of results. The level of precision of the presented results has also been reduced. Finally, we have revised section 3.6 to a comprehensive discussion section (Section 5) and moved all discussion here, which should considerably reduce the repetition identified by the reviewer.**

Specific comments:

*Line 60: ozone change also has a climate impact – a bit strange to only speak of water vapor as having a climate effect.*

**In the revised version several changes have been made to the introduction to more accurately represent the climate effects of supersonic emissions. In particular, it is clarified that the changes in $O_3$ affect global RF and the role of aerosol emissions has also been described (Lines 5–61).**

*Line 66-68: this is the case also for aviation in general, not just for a supersonic fleet – may want to make that point.*

**The suggested addition has been implemented in the revised version (lines 69-71).**

*Line 70: use response instead of impact or specific impact of what*

**The line has been adjusted to use "response" (line 72).**

*Line 84-86: it's a bit unclear from this and the preceding sentence how you evaluate an improvement – please could you specify or use comparable units for the two set of numbers reported*

**The "improvement" referred to a reduction of the variability between the models relative to the model-mean result, which we understand is not clear form the text.**

**As discussed in the general comments of RC2, we have reduced the discussion of modelling "improvements" in the revision. The mention of improvement has been**

**removed entirely, instead we only highlight the variability of the model results compared to the mean to demonstrate inter-model variability (Lines 86-89).**

*Line 105: should be clear about why 2050 atmosphere means – as far as I can tell, there is not change in meteorological/climatic conditions to align with a projected SSP370 climate. Also, the description of what is done varies between models – for instance MOZART uses RCP 4.5, do all the models change e.g. CH4 concentration to SSP370 levels, GEOS-Chem only mentions volcanic emissions (what do other models do?), etc.*

**The reviewer's feedback has been incorporated into an overall revision of the methodology (now section 3, "Atmospheric Modelling"). The model descriptions and structure thereof is standardized and streamlined, adding the missing descriptions identified in this comment as well. The new introduction to this section now better clarifies what we mean with a 2050 atmosphere, and how the implementation thereof is handled between the models (Lines 135-141).**

*Line 110: it's not quite clear to me why – is it because the implementation of supersonic aircraft happens on a different baseline aviation sector, i.e. SSP370? Would be good to clarify.*

**As a baseline we use SCENIC emission scenarios from Grewe and Stenke (2008), which differ slightly from the scenarios in Grewe et al. (2007) as they account for the partial replacement of subsonic traffic by the supersonic aircraft. This leads to the ~3.3% reduction in global fuel consumption compared to the Grewe et al. (2007) inventories. Given the similarity of our table 1 to the table in Grewe et al. (2007), we felt that this needed clarification to explain why these numbers do not match exactly.**

**We see that this only addition may only cause confusion instead, therefore we have revised the scenario descriptions to be more to-the-point and removed the unnecessary comparison with the Grewe et al. (2007) paper (Lines 110-121).**

*Table 1: a bit unclear. The left-hand side says "all aircraft" and the right "supersonic aircraft" but looking at it, it doesn't seem like the all aircraft fuel consumption in the different scenarios adds up to the A0 plus whatever the introduction of a supersonic fleet does – which I'd expect from the titles. Please clarify.*

**The different scenarios don't add up because of the partial replacement of subsonic traffic by supersonic aviation, which reduces the subsonic fuel consumption compared to the baseline scenario.**

**We have fully revised table 1 to be more concise, with a clear split between subsonic and supersonic aircraft emission characteristics instead. This should communicate the differences much more clearly.**

*Line 215: what exactly does fixing in the troposphere mean?*

Here "fixing" referred to the use of prescribed mixing ratios for tropospheric $H_2O$. We realize this is not clear, and the terminology has been revised to describe that the models use prescribed mixing ratios instead (Lines 233-234).

*Line 215-216: this seems like a quite important problem, making me question inclusion of this the model in the model-mean values reported at this stage. But maybe the authors have more confidence than is reflected by this wording – if so, I would suggest modifying*

We agree with the reviewer's reservation. Upon further investigation when processing the new RF calculations, we found that there was a bug in the mass calculations which inadvertently doubled the $H_2O$ perturbation mass. The presented results and manuscript have been updated with new values for the perturbation masses for MOZART-3, and its results now fit in with that of the other models. The line in question, and other parts discussing the initially theorized problems with the $H_2O$ modelling in MOZART-3 have been removed. We now have no reservations regarding the inclusion of the MOZART-3 results.

*Line 226: again, unclear what a 2050 atmosphere means*

The line in question has been removed from the manuscript, per the previously discussed revision of the methodology section.

*Line 248: missing a +-sign? Also, text should specify what the percent number is relative to.*

The + was missing, but in the revised version + notations have been removed from positive numbers entirely. The line marked by the reviewer has been removed in the revision.

*Line 253: rather, the change in emission cause*

This was overlooked in the wording and has been amended (Line 275).

*Line 260: without any quantitative measure or validation, and given the caveats and differences in model experimental setup listed above, I question whether you can and should confidently say that reduced model spread is a "considerable improvement". If Suggest rephrasing to "considerably lower model spread" or "closer model agreement" Importantly, I do not agree that the model agreement seems to be smaller when looking at the spatially resolved figures. So this needs to be said in a more nuanced way.*

We agree. The statement in question is omitted in the revision, and the content is moved to a paragraph in the discussion. This only discusses that the spread is smaller instead (Lines 562-572).

*Line 265: is 3.4 the model average? Not clear from the text/table*

This is indeed the model-mean, the text has been updated to clarify (Lines 285).

*Table 3: for ozone, is it the full column or stratospheric only?*

**We present changes full column ozone, table 3 has been updated to improve clarity on which variables are stratospheric and global.**

*Line 277: can the authors explain why this occurs?*

**As also described in the similar question of RC1, during the revision we found that this result was inadvertently calculated from 3 years of EMAC data. In the revised version we calculate the mass perturbation from all 6 years instead, which no longer yields a net-decrease in southern hemispheric $H_2O$. We therefore expect this was the result of the model's internal variability, although the EMAC response does still indicate regions of local decreases of water vapour near the southern tropopause, but we have been unable to pinpoint exact explanations for this.**

**The line in question is removed.**

*Fig 2: caption says in response to supersonic emissions scenario (A0) – should this be A1?*

**This should indeed be the case, thank you. Please note that in the revision we changed the notation of scenarios to S0 to S4 to avoid confusion with appendices.**

*Line 294-295: what does extended vertical domain mean? That the models have a higher upper model layer? This seems an important bit of information when looking at a perturbation to the atmosphere that is so altitude dependent and some more detail would be helpful.*

**This does indeed refer to a higher upper model layer and the higher vertical resolution of the models.**

**The text has been amended with more detail (Line 563-564 in the revised discussion). Furthermore, per our response to RC1 the manuscript has been amended with further comparisons of the vertical model grids and discussion thereof (Figure A1, Lines 160-161, 188, 216, 238, and 327-350).**

*Line 306: please elaborate on the term "tropical pipe"*

**The term "tropical pipe" refers to the main region of upwelling over the tropics.**

**We have clarified this in the text (Line *368*)**

*Line 311: is discrepancy the best word for describing model differences? Requires a baseline to relate it to? Maybe large spread instead*

**The wording of this line has been altered to use "spread" instead of discrepancy (Line 327).**

*Line 396: not sure I'm convinced that EMAC responses can be said to be similar here, moreover, when looking at the vertical profiles there seems to be neither similarity nor substantially lower model spread than in the Grewe et al study. This applies to other places as well, where the word similar is used for the model comparison.*

**We have revised this line, separating the comparison between the offline models from EMAC. This more accurately reflects the contrast of the EMAC results with the other models (Lines 402-408).**

**Elsewhere in the manuscript, the wording is updated to clarify that the spread is reduced in terms of the ozone perturbation masses only (Lines 565-567), and comparisons of this sort with the Grewe et al. study have been reduced.**

*Line 398-399: I did not readily find out how the different models treat heterogenous chemistry, some have polar stratospheric clouds as well – some more detail for all models in the methods section would be helpful. Do all four models have aerosol treatment included?*

**In the revision of the methodology we have ensured that aerosol modelling is described for each of the models, and that it is mentioned explicitly if aerosols are not modelled (Lines 149-150, 180-183, 203-206, 230-233). This is repeated in the revised results, given the importance of this characteristic (Lines 445-446).**

*Line 401: should be ozone layer's self-healing effect? Not sure what is means for ozone to be self-healing... Also, what are smog forming processes when you talk about lower-stratosphere? I think it's better and more clear to describe in terms of chemistry. And relate it to the NOx emission changes displayed in Fig1.*

**The wording has been adjusted. The mention of "smog processes" has also been altered to use "$NO_x$-driven ozone formation" instead (Line 405).**

*Line 417-418: the cited study is very old and much older than the Grewe et al study. Are there more recent model documentation or evaluation papers that can be used to support this statement? Alternatively, what improvements have been made to these models compared to the ones used in Grewe et al.*

**In the light of the previously mentioned revisions and the general reviewer comments, we have removed almost all direct comparison of our models and those used by Grewe et al. (2007) in the revision. This line in question has also been removed.**

**References:**

Grewe, V., Stenke, A., Ponater, M., Sausen, R., Pitari, G., Iachetti, D., Rogers, H., Dessens, O., Pyle, J., Isaksen, I.S.A., Gulstad, L., Søvde, O.A., Marizy, C., Pascuillo, E., 2007. Climate impact of supersonic air traffic: an approach to optimize a potential future supersonic fleet – results from the EU-project SCENIC. Atmospheric Chemistry and Physics 7, 5129–5145. https://doi.org/10.5194/acp-7-5129-2007